Resource

# Three million images and morphological profiles of cells treated with matched chemical and genetic perturbations

Srinivas Niranj Chandrasekaran[1], Beth A. Cimini [1], Amy Goodale[1], Lisa Miller[1], Maria Kost-Alimova[1], Nasim Jamali[1], John G. Doench [1], Briana Fritchman[1], Adam Skepner [1], Michelle Melanson[1], Alexandr A. Kalinin [1], John Arevalo [1], Marzieh Haghighi [1], Juan C. Caicedo[1], Daniel Kuhn [2], Desiree Hernandez[1], James Berstler[1], Hamdah Shafqat-Abbasi[1], David E. Root [1], Susanne E. Swalley[3], Sakshi Garg[2], Shantanu Singh[1] ✉ & Anne E. Carpenter [1] ✉

The identification of genetic and chemical perturbations with similar impacts on cell morphology can elucidate compounds' mechanisms of action or novel regulators of genetic pathways. Research on methods for identifying such similarities has lagged due to a lack of carefully designed and well-annotated image sets of cells treated with chemical and genetic perturbations. Here we create such a Resource dataset, CPJUMP1, in which each perturbed gene's product is a known target of at least two chemical compounds in the dataset. We systematically explore the directionality of correlations among perturbations that target the same protein encoded by a given gene, and we find that identifying matches between chemical and genetic perturbations is a challenging task. Our dataset and baseline analyses provide a benchmark for evaluating methods that measure perturbation similarities and impact, and more generally, learn effective representations of cellular state from microscopy images. Such advancements would accelerate the applications of image-based profiling of cellular states, such as uncovering drug mode of action or probing functional genomics.

Image-based profiling of cell samples is proving increasingly useful for biological discovery[1]. In image-based profiling, cells are treated with perturbations of interest and the resulting morphology is captured by microscopy. Cell morphology is quantitatively compared with identify meaningful similarities and differences among the perturbations, in the same way that transcriptional profiles are used to compare samples. More than a dozen applications have been demonstrated[1], including identification of the mechanisms of a disease by comparing cells from patients with a disease to those of healthy patients, identification of the impact of a chemical compound by comparing cells treated with it to untreated cells, and identification of gene functions by clustering large sets of genetically perturbed samples to determine relationships among the genes.

Typically, morphological features are extracted from each cell using classical image processing software. These so-called 'hand-engineered' features have been carefully developed and optimized to capture cellular morphology variations, including size, shape, intensity and texture of the various stains in the image. These features are the current standard in the field and require post-processing steps including normalization, feature selection and dimensionality

[1]Broad Institute of MIT and Harvard, Cambridge, MA, USA. [2]Merck Healthcare KGaA, Darmstadt, Germany. [3]Biogen Inc., Cambridge, MA, USA. ✉e-mail: shantanu@broadinstitute.org; anne@broadinstitute.org

**Fig. 1 | Sample images from the dataset.** A five-channel image of human U2OS cells treated with the compound PFI-1 (a BRD4-specific inhibitor). This is a representative image from one of four wells of cells treated with PFI-1. The channel names indicate the cellular structures identified in each image (see Methods section for details; AGP, actin, Golgi, plasma membrane; DNA, nucleus; ER, endoplasmic reticulum; mito, mitochondria; RNA, nucleoli and cytoplasmic RNA). Other example images (including brightfield channels not shown here) are available at https://github.com/jump-cellpainting/2024_Chandrasekaran_NatureMethods/tree/6ba3fcd1495d9e844e4607373a568641981ffcd8/example_images. Scale bar, 100 μm.

reduction[2]. With advances in representation learning during the last decade, it is natural to ask what set of features could be automatically identified by machine learning algorithms, directly from pixels.

However, image-based profiling has yet to fully benefit from the latest machine learning research. The vast majority of studies use classical segmentation and feature extraction; deep learning methods are beginning to be explored and there is much room for advancement[3,4]. Historically, the lack of ground truth has been a major limiting factor in the field; that is, the true relationships among perturbations (for example, genes and compounds) are unknown and require significant effort to ascertain[5]. Although this is exciting because the potential for biological discovery is high, the lack of ground truth presents a challenge for optimizing deep learning pipelines.

Here we describe our design and creation of a benchmark dataset via a single large experiment, CPJUMP1. The dataset consists of approximately 3 million images of cells, image-based profiles of 75 million single cells, and well-level aggregated profiles. A sample five-channel image is shown in Fig. 1. This dataset contains chemical and genetic perturbation pairs that target the same genes and are tested in separate wells to see whether they produce similar (or opposite) phenotypes. Although these pairings are not absolute truth for a number of reasons discussed later, they are nevertheless more likely than random pairs to match (that is, induce similar or opposite morphology changes). This Resource is unique because there are no other image-based datasets of the Cell Painting assay (described later) that include pairs of annotated genetic and chemical perturbations performed side by side, under different experimental conditions such as different cell types, time points and imaging conditions. These were also executed in parallel to minimize technical variations that may confound the signal. There are many public Cell Painting datasets (for example, https://github.com/broadinstitute/cellpainting-gallery) but we are aware only of one with genetic and chemical perturbation types run in parallel (RxRx3, https://www.rxrx.ai/rxrx3). RxRx3 has not been provided with gene–compound relationship annotations and all but 733 genes are anonymized; any pairs that exist would be scattered across many plates and batches.

As well, it includes only a single cell type, time point and imaging condition. Thus, the nature of CPJUMP1 enables the testing of computational strategies to optimally represent the samples so that they can be compared and thus uncover valuable biological relationships. It also enables comparison of CRISPR-Cas9 knockout and ORF (open reading frame) overexpression as mechanisms to perturb genetic pathways and to identify the compounds' mechanisms of action.

## Results

To push forward advancements in this field, we assembled a consortium of 10 pharmaceutical companies, two non-profit institutions, and several supporting companies, known as the JUMP Cell Painting Consortium (Joint Undertaking in Morphological Profiling). Members of this Consortium created the ground truth dataset we present here, for optimizing the main assay used in image-based profiling, called the Cell Painting assay, and to move methods in the field forward[6,7]. We selected and curated a set of 160 genes and 303 compounds with (relatively) known relationships between each other, and designed an experimental layout to enable testing and comparing methods to quantify their similarities (Methods), all with a strong emphasis on making this dataset useful for developing computational methods for the field.

There are two groups of experimental conditions in this dataset, the primary group and the secondary group. In the primary group we separately captured chemical and genetic perturbation (CRISPR knockout and ORF overexpression) profiles in two cell types (U2OS and A549) at two time points (Supplementary Table 1; a representation of profiles from a subset of this experiment is shown in Fig. 2). There are a total of 40 384-well plates in the primary group of experimental conditions (Fig. 3). The secondary group consists of additional plates of experimental conditions as well as plates from the primary group that have undergone additional imaging conditions (Fig. 3), which are described in the Methods section. In addition to being used to optimize the assay conditions[7], the primary and secondary groups offer multiple views of cells treated with each chemical or genetic perturbation, and therefore can be used for many interesting machine

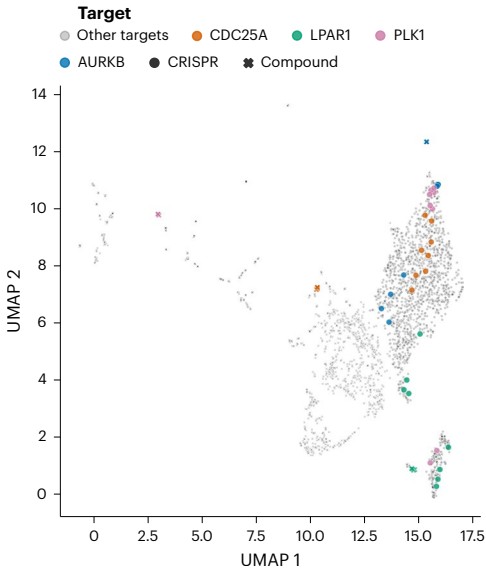

**Fig. 2 | UMAP representation of image-based profiles from a subset of the primary group of tested samples.** Profiles from human A549 cells perturbed by all compounds and CRISPR guides at the long time points (Supplementary Table 1) are shown here. The top four most-similar compound–genetic perturbation pairs, associated with the same gene, are highlighted.

learning explorations, such as style transfer (for example, to attempt prediction of one experimental condition from another), information retrieval and multi-view learning, and for benchmarking representation learning methods. There are 11 additional plates in the secondary group, but there are 67 plates of images because several plates were imaged multiple times (Fig. 3; Methods).

A major goal of image-based profiling is to derive a representation from the quantitative images of cell samples, such that samples in biologically similar states have similar representations. Given such a representation, solutions for many of the applications discussed become immediately feasible. Here we benchmark representation learning as a foundation for methods development in the future.

As a way to compare different representation methods, we created benchmarks based on two tasks: detection of differences between perturbations and negative controls to identify active perturbations, and the grouping of gene–compound pairs in which the gene's product (protein) is a target of the compound (as well as grouping two CRISPR guides targeting the same gene, or two compounds annotated with the same target). For both tasks we use cosine similarity (or its absolute value), a simple but widely used correlation-like metric, to measure similarities between pairs of well-level aggregated profiles. In some cases the expected directionality of correlation is positive while in other tasks the correlations may be strongly positive or negative; we adjust statistical tests for each task accordingly.

### Benchmarking perturbation detection methods

We chose perturbation detection as one of the tasks to evaluate representations because it often precedes other useful applications (by removing samples that have no or little true signal), and is equivalent to measuring statistical significance of the perturbation's signal. For example, a set of chemical or genetic perturbations might be filtered by this phenotypic activity criterion before embarking on subsequent laboratory experiments, or prior to training a model, or other analysis that could be confounded by noisy signals. It can also be useful for determining which experimental protocol or computational analysis pipeline is most sensitive from among several alternatives. It should be noted that even given perfect computational methods for

feature extraction, batch correction and profile comparison, many samples will be detectably different from negative controls for several biological reasons. For example, a chemical or genetic perturbation may affect cell morphology only in a particular cell type, under particular environmental conditions, at a particular time, or if particular stains were used, conditions that may not have been met in the experiment. Conversely, a perturbation's impact may be amplified by the plate layout, given that even unrelated perturbations in the same well position might look similar. This concern is overcome by matching treatments in different well positions, where such data are available (see Benchmarking perturbation matching methods).

We used average precision to measure each primary group sample's ability to retrieve its replicates against the background of negative control samples, using cosine similarity as the similarity metric. The significance of the average precision value is assessed using permutation testing to obtain a $P$ value, which is then adjusted using the false discovery rate to yield a corrected $P$ value ($q$ value). We calculate the mean of the average precision for each perturbation and then term the fraction of perturbations with a $q$ value below the significance threshold (0.05) as the fraction retrieved. Details about the computation of average precision and fraction retrieved are provided in the Methods section.

In general, we find that the fraction retrieved for compounds is higher than that of genetic perturbations, across all conditions (Fig. 4a). This indicates that chemical compounds produce phenotypes that are more distinguishable from negative controls, compared with phenotypes produced by CRISPR knockout and ORF overexpression. We also find that the fraction retrieved is higher for CRISPR knockout than for ORF overexpression (Fig. 4a). In summary, compounds, CRISPRs and ORFs all yield signals in the assay, with the compounds being the strongest and ORFs the weakest. However, we emphasize a strong technical variable that precludes a strong conclusion here: the reduced fraction retrieved values for ORF may be attributed to plate layout effects, in which identical treatments in different rows or columns have dissimilar profiles. This factor amplifies the systematic technical noise in the compound and CRISPR plates due to their particular layout, while it adversely impacts ORFs (Methods). Retrieving the same position replicates for ORF does increase the fraction retrieved, as would be expected if plate layout effects are substantial (Extended Data Fig. 1). Plate layout effects may be partly mitigated by mean centering every feature at each well position, although we have not applied the correction to this dataset because we do not have sufficient diversity of samples in each well position across a large number of plates. Furthermore, although there is a presence of signal for all three perturbation modalities, we note that these were not random sets of genes and compounds; instead, the compounds were chosen from the Drug Repurposing set[8] and the genes were selected secondarily depending on whether their protein products are targeted by those compounds. This is likely to select for reagents that yield phenotypes versus random genes and compounds.

### Benchmarking perturbation matching methods

We next established a benchmark for researchers to develop and test strategies for a real-world retrieval task, in which we search for genes or compounds that have a similar impact on cell morphologies as the query gene or compound. Improved methods would enable improved discovery of the compounds' mechanisms of action based on a compound query[9], and virtual screening for useful compounds based on a gene query[10]. This dataset presents a unique opportunity to match profiles of perturbations across modalities (chemical versus genetic) because genes in this dataset are targeted by two types of genetic perturbations (ORF and CRISPR-Cas9 knockout) and by at least two compounds. Similarly, because there are both a pair of CRISPR guides and a pair of compounds targeting each gene or gene's product, this dataset can be used to match profiles within a perturbation modality (there is only one ORF reagent per gene, therefore a similar analysis is

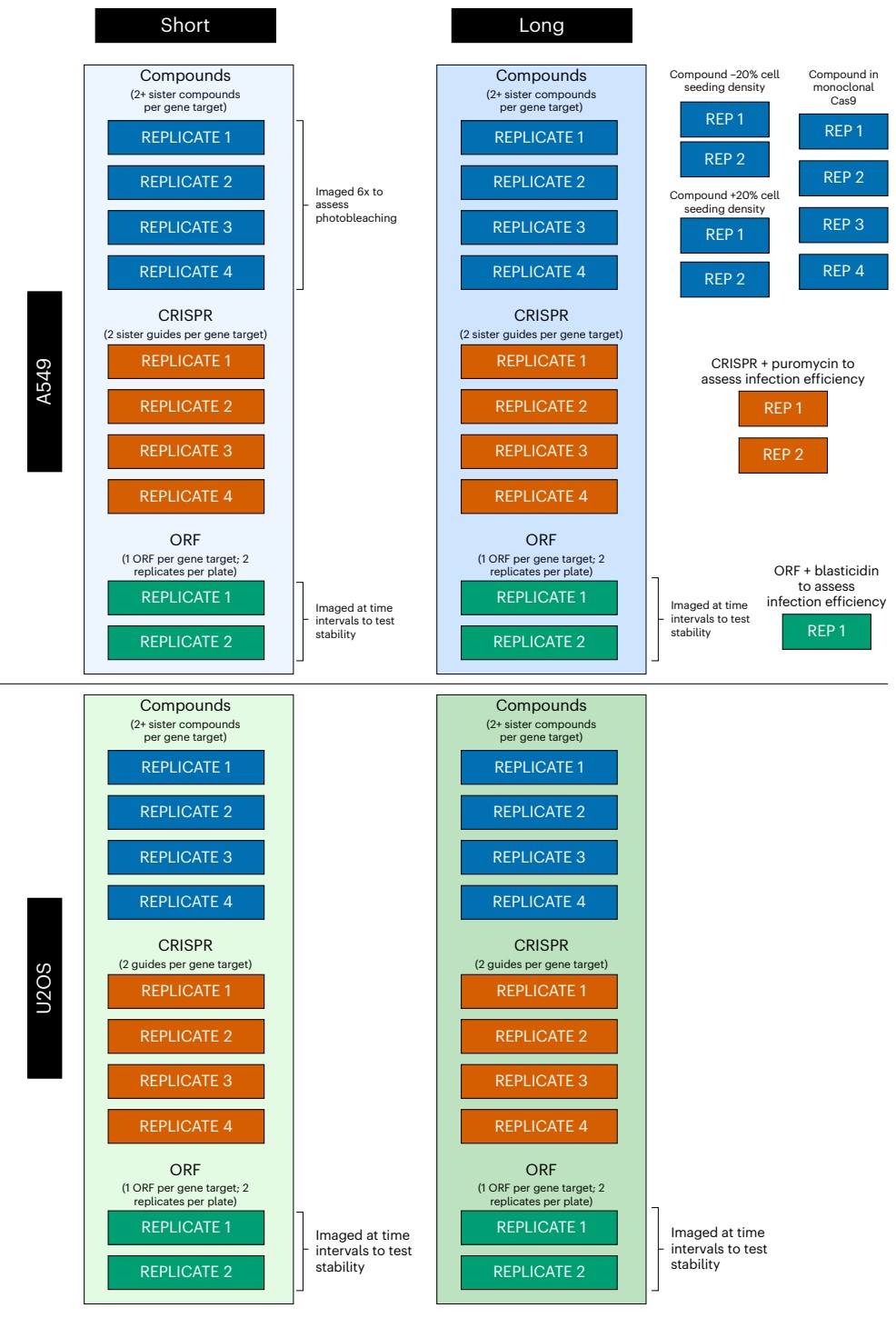

**Fig. 3 | Schema of the data generated in the CPJUMP1 experiment.** Each rectangular block of replicates (REPLICATE X or REP X) is a 384-well plate of cells perturbed by compounds, CRISPR guides or ORFs and subjected to different experimental conditions. Short and long time points are described in Supplementary Table 1. Plates in the vertical blue and green boxes comprise the primary CPJUMP1 experiment. The other (that is, secondary) experimental conditions are described in the Methods section.

not possible for overexpression). It also offers an opportunity to study the directionality of profile matching; for example, whether CRISPR knockouts and ORF overexpressions consistently yield anti-correlated profiles.

After filtering out perturbations that were indistinguishable from negative controls ($q > 0.05$), we then evaluated average precision to identify true connections (that is, perturbation pairs that target the same gene or gene's product), which are distinct from false connections (that is, pairs not known to target the same gene or gene's product).

We first tested the ability to retrieve true connections within the same perturbation modality: that is, 'sister' compounds that are annotated as targeting the same gene should match each other, and 'sister' CRISPR guides that target the same gene should match each other. Because compounds can enhance or inhibit the function of a protein (or have other impacts), those with the same gene annotation might be positively correlated or negatively correlated; for compounds, we therefore used the absolute value of cosine similarities while calculating average precision. By contrast, for CRISPR guides, those targeting

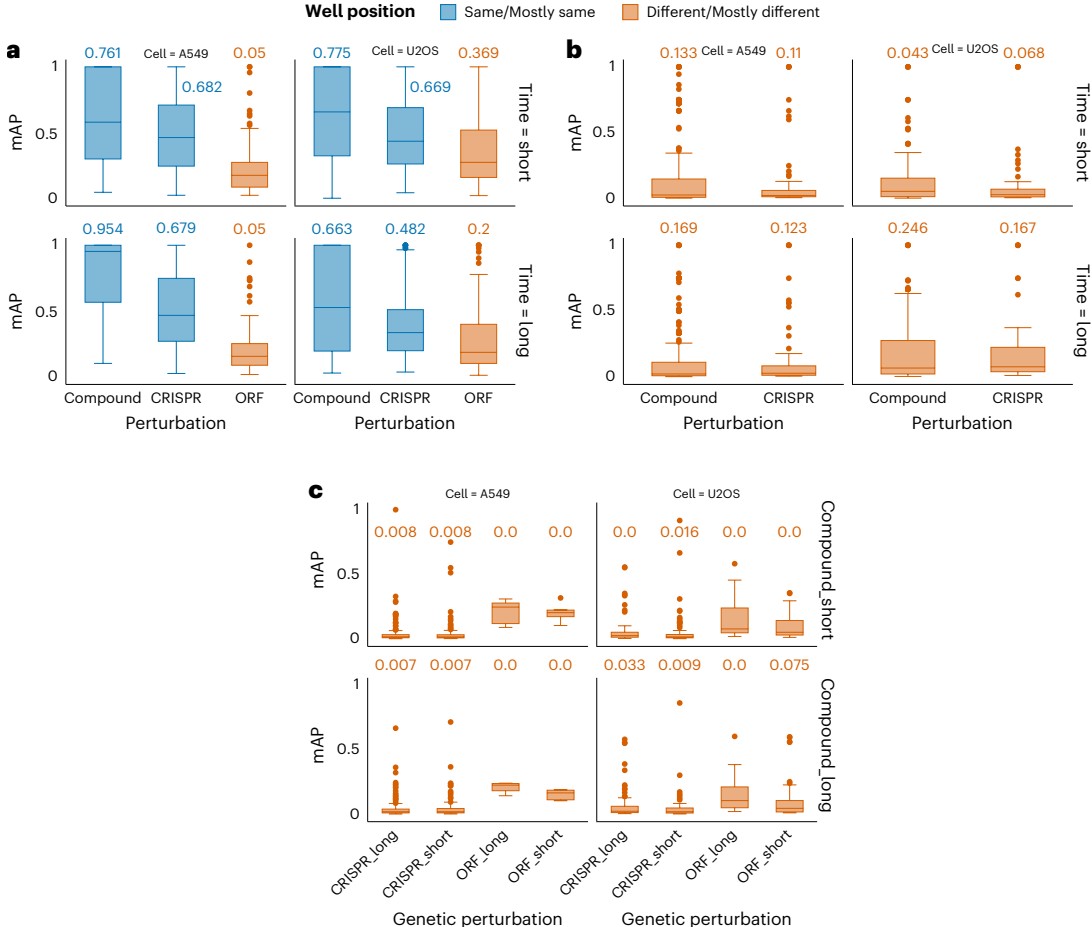

**Fig. 4 | Benchmark results for progressively more difficult retrieval tasks.**
**a**, Perturbation detection: retrieving replicates of the same sample. Mean average precision (mAP) for perturbation detection is shown across experimental conditions: cell type (columns) and time points (rows; short and long time points are defined in Supplementary Table 1). The numerical values shown above each box plot are the fraction of perturbations that can be successfully retrieved for each retrieval task. Box plot boundaries are 75th (Q3) and 25th (Q1) percentiles, with whiskers at ±1.5-fold the interquartile range (Q3–Q1) or the highest or lowest data point. The color of the bars denotes whether the query perturbation and the retrieved perturbation are in mostly the same well position (blue) or mostly different well positions (red); the latter is a more challenging task due to technical well-position artifacts. **b**, Perturbation matching, within a perturbation type: the plot shows mAP for sister perturbation retrieval (that is, pairs of compounds or pairs of CRISPR guides annotated with the same gene target). ORFs are not shown because there is only a single ORF reagent per gene. Absolute cosine similarity is used for calculating mAP values for compounds because pairs of compounds annotated to target the same protein can be positively or negatively correlated. **c**, Perturbation matching, across perturbation types: the plot shows mAP values for retrieving compound–gene pairs (that is, the same target and different perturbation type). Absolute cosine similarity is used for calculating mAP values for both compound–CRISPR and compound–ORF matching. The number of independent biological samples is available in Supplementary Table 2.

the same gene are expected to be positively correlated, therefore we used the actual values of cosine similarities, as in the perturbation detection task.

With baseline methods, this task is (not surprisingly) much more challenging than retrieving replicates of the same sample (compare Fig. 4a,b). Of all of the compounds that yield a signal, only ~5–25% of them correctly match their sister compounds targeting the same protein. Likewise, and more surprisingly given their expected accurate annotation and specificity, only 7–17% of the CRISPR reagents correctly match to their sister guides targeting the same gene. We cannot distinguish the many factors that are likely to make this a challenging task, including non-optimal ground truth annotations for compounds, off-target effects (for compounds and CRISPR guides), differing levels of knockdown for CRISPR guides, lack of information content in the assay, polypharmacology in which each compound impacts multiple targets (see Discussion) and/or non-optimal methods for matching samples. Although the values of fraction retrieved are similar for compounds and CRISPR guides, more compounds are distinguishable from negative control than CRISPR guides (Fig. 4a and Supplementary Table 2). Thus, surprisingly, retrieving sister compounds is more

successful than retrieving sister CRISPR guides (Fig. 4b), perhaps because compounds tend to induce stronger phenotypes (Fig. 4a).

Next, we assessed cross-modality matching: that is, the ability to retrieve correct gene–compound pairs. Retrieving compound–gene pairs is more difficult than perturbation detection and sister perturbation retrieval (which itself reached only ~25% in the best scenario), but it is extraordinarily useful for identifying novel chemical regulators of genes and identifying the mechanism of query compounds. Even a low success rate, therefore, can accelerate drug discovery. Given the potential for compound–gene matches to be positive or negatively correlated (detailed in the next paragraph), we used the absolute cosine similarities for both compound–ORF and compound–CRISPR retrieval.

Gene–compound retrieval is only slightly better than expected by chance (Fig. 4c) and, as expected, is less effective than compound–compound and CRISPR–CRISPR matching (Fig. 4b). This might reflect that gene–compound annotations are less reliable than compound–compound annotations (and certainly less reliable than CRISPR guide annotations, for which the target gene is designed a priori to be accurate and specific), and/or that better methods are needed to align data across modalities. Comparing the two genetic perturbation

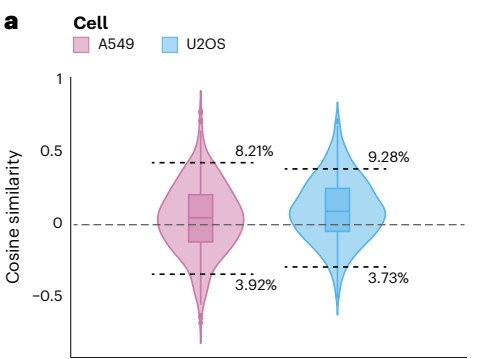

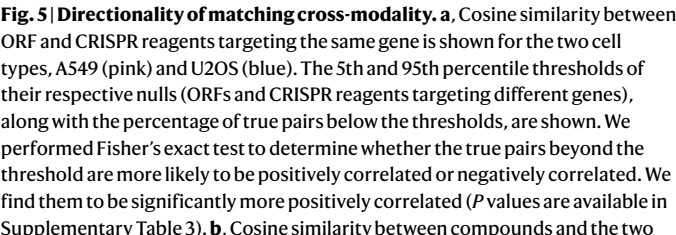

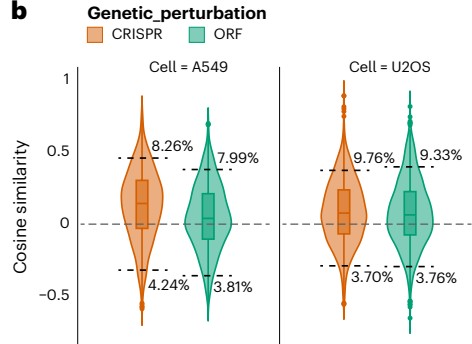

**Fig. 5 | Directionality of matching cross-modality. a**, Cosine similarity between ORF and CRISPR reagents targeting the same gene is shown for the two cell types, A549 (pink) and U2OS (blue). The 5th and 95th percentile thresholds of their respective nulls (ORFs and CRISPR reagents targeting different genes), along with the percentage of true pairs below the thresholds, are shown. We performed Fisher's exact test to determine whether the true pairs beyond the threshold are more likely to be positively correlated or negatively correlated. We find them to be significantly more positively correlated (*P* values are available in Supplementary Table 3). **b**, Cosine similarity between compounds and the two genetic perturbation modalities, CRISPR (orange) and ORF (green) targeting the same gene or gene product. All analyses here were also statistically significantly more positively correlated; *P* values are available in Supplementary Table 4. Cosine similarity of zero is shown as a gray dashed line in both subplots. In **a** there were $n$ = 3,728 biological independent ORF and CRISPR reagents and in **b** there were $n$ = 1,864 independent pairs of compounds and genetic perturbations targeting the same gene or gene product. Box plot boundaries are 75th (Q3) and 25th (Q1) percentiles, with whiskers at ±1.5-fold the interquartile range (Q3–Q1) or the highest or lowest data point.

modalities, we find that the values of mean average precision for retrieving compound–CRISPR pairs were better than that of compound–ORF pairs across various cell types and time points, except at one time point, where ORFs match better to compounds than CRISPR reagents do (Fig. 4c).

Although the performance for compound–genetic perturbation retrieval is low compared with the other retrieval tasks discussed above, it should be strongly noted that significant time and resources are otherwise required to identify the target of a compound, and similarly to identify compounds that target a particular gene. Therefore, even the baseline's relatively low matching rates might accelerate drug development by yielding a list of possibilities for biologists to test directly in subsequent experiments, or to be combined with orthogonal lists of candidate targets for a compound, to improve accuracy. Improving image representations and thus the accuracy of predicted matches by a few percentage points could therefore have a major impact on the discovery of compounds that impact proteins of interest, and the identification of the mechanism of action of compounds of interest.

Finally, we examined the directionality of gene–compound matching, that is, whether a compound targeting a protein encoded by a given gene has a correlating or anti-correlating profile with CRISPR (which reduces the amount of gene product) and with ORF (which increases the amount of gene product). Most compounds are annotated as inhibiting the function of their target gene's product, therefore one might expect image-based profiles from cells treated with CRISPR guides to generally positively correlate to (mimic) the corresponding compound's profile, whereas ORF profiles might generally be expected to anti-correlate (oppose) the corresponding small molecule's profile because overexpression often increases a gene's function. By the same rationale, ORFs and CRISPR guides targeting the same gene might be expected to yield opposite (anti-correlated) effects on the cells' profiles. However, we strongly note that there will be numerous exceptions, given the nonlinear behavior of many biological systems, and any number of distinct mechanisms for which these general principles may not hold, which we have previously detailed[10]. For example, many compounds do not inhibit their target protein's function but instead activate it or induce some new function, and many overexpressed genes may have no impact at all, or even have a dominant negative or feedback loop or compensatory impact on the gene's function. Furthermore, the choice of cell type, time point or readout for capturing the similarity may not be optimal. In fact, the exceptions

may be more common than the commonsense rules in this case. One aim of generating this dataset was to quantify how often the expected relationships and directionalities occur, to provide concrete evidence in this so-far theoretical debate with only anecdotal evidence available.

We began by testing the basic hypothesis that CRISPR and ORF reagents targeting the same gene should yield negatively correlated (opposite) profiles to each other. Surprisingly, we found that the CRISPR and ORF profiles are slightly positively correlated with each other, in both cell types (Fig. 5a and Supplementary Table 3). We then compared the cosine similarities between compound–CRISPR pairs and compound–ORF pairs that target the same gene. In both U2OS and A549 cells, we found that both compound–CRISPR and compound–ORF pairs were more positively correlated than negatively correlated (Fig. 5b and Supplementary Table 4).

Looking at the strongest-matching positive and negative gene–compound pairings (Supplementary Tables 5 and 6), we found many pairings with explainable directionality (for example, CRISPR knockout of a gene matches a compound annotated as inhibiting the protein product of that gene). For example, the top positively correlated gene–compound match in U2OS cells is the PLK1 inhibitor compound BI-2536 matched with CRISPR against PLK1 (Extended Data Fig. 2), and the next two matches are annotated as Aurora kinase inhibitors that match CRISPR against AURKB (Supplementary Table 5). Similarly, the five strongest matches in A549 cells (Supplementary Table 6) are all CRISPR reagents positively correlating with compounds annotated as targeting the correct protein encoded by a given gene. Some overexpressions also match the expected directionality, such as the second strongest negatively correlated match in U2OS cells, which is the compound GSK2110183, an AKT inhibitor negatively correlated to overexpression of AKT1. Still, there were many pairings with unexpected directionality, possibly due to ORFs exhibiting dominant-negative behavior, as in the seventh positively correlated gene–compound match in U2OS cells, the compound pentostatin, which is annotated as an adenosine deaminase inhibitor; it matches with overexpression of ADA (adenosine deaminase). Unexpected directionality is evident when examining BRD4 overexpression and BI-2536 treatment, both of which induce cell death (Extended Data Fig. 2). Because BI-2536 inhibits BRD4, one might expect BRD4 knockout to closely align with BI-2536's effects. This discrepancy suggests two possibilities: either the multi-target nature of BI-2536 leads to a dominant phenotype (PLK1 inhibition may be stronger than BRD4 inhibition), or BRD4 inhibition

has a limited phenotypic impact. This latter possibility aligns with the negative control-like phenotype observed with the BRD4-specific inhibitor, PFI-1 (Extended Data Fig. 2). Some CRISPR reagents produce surprising directionality, as in the case of the compound TG-003 in U2OS cells, annotated as a CLK inhibitor but negatively correlated to CRISPR knockout of CLK1. Given these findings, and the possibility that compounds often behave differently in different cellular contexts and may be annotated based on a particular one, it is not surprising that compounds targeting the products of particular genes do not show a consistent directionality relationship with ORFs or CRISPRs.

## Discussion

Biology would benefit greatly from the machine learning community turning its attention to rich, single-cell imaging data. Although our results may only touch upon the potential applications of machine learning, our emphasis is a strategic appeal to the machine learning community. We hope that the Resource and benchmarks we have created will provide a foundation on which researchers can develop and test novel methods for representation learning, multi-view learning, information retrieval and style transfer, among others. The task of identifying targets of a compound to understand its mechanism is exceptionally difficult, expensive and time-consuming, creating a major bottleneck in developing useful compounds for chemical biology and drug discovery[11]. This has several implications for this work: first, even with low but non-zero success rates, biologists can use their understanding of the compound's known traits to create and test hypotheses about a chemical's mechanism of action, accelerating discovery. Second, the predictions from this method might be combined with other predictive approaches, such as rank ordered candidate gene lists from structure-based chemical–protein binding predictions, to produce more reliable results. And last, we also emphasize that, as a direct result of the fact that identifying the targets of compounds is so difficult, only a very limited amount of rather noisy ground truth exists; each known compound–gene interaction has been painstakingly discovered after hundreds of thousands of dollars of effort over many years, and many pairings are uncertain. By contrast, many mainstream machine learning tasks are oriented to replicate specific human skills where ground truth can be collected at large scale (for example, translation or image recognition), and, given sufficient resources, the accuracy often approaches 100%. We believe that supervised methods hold promise[1], and we hope that novel machine learning methods developed using our dataset will be used to discover new gene–compound connections that can add to the ground truth for this problem in the future.

Beyond biology, our dataset provides a challenging, real-world test bed for many kinds of more general machine learning algorithms. It is a large-scale perturbation experiment with complex multidimensional, hierarchical data (images displaying dozens of cells each), and we believe that new machine learning strategies still need to be developed to realize its full potential. In addition to the prediction problems we present in this paper, it also opens up problems in high-level reasoning on experimental data, enabling the study of complex artificial intelligence strategies, such as causal inference (observations from interventional experiments), planning (optimizing the next intervention that maximizes discovery), and simulations (what would have happened if other interventions are applied).

Finally, unusual aspects of the data type that we present pose challenges to machine learning algorithms and will require that they be pushed in different directions to adapt. This may spark creative solutions with broader impact in machine learning. For example, multiplexed imaging will push the field of machine learning to adapt to domains outside the red–green–blue (RGB) colorspace of natural images in which the number and relationship between the channels (for example, the extent of correlation) is very different, compared with natural images of everyday objects.

In general, deep learning-based features may provide improvements in performance for some tasks over the classical CellProfiler-based features, although interpretability remains a challenge. The direct interpretability of CellProfiler-based features can help in examining signatures and comparing them to evidence found in existing research relevant to a particular profiling task. One can readily visualize any target cell population by looking up the associated sample cell images or by converting CellProfiler representations into images with convolutional neural network-based image generators. An example set of cell images generated by a fundamental version of such a model is shown in Extended Data Fig. 3. The research community can delve deeper into creating interpretative models from CellProfiler profiles that we have provided in this resource.

Still, this dataset has limitations. It covers only 160 genes and 303 compounds, and ~21% of compounds in this dataset are annotated as targeting proteins in more than one gene family (Supplementary Table 7). Ideally, such a dataset would include only compounds that are very well-studied as targeting one and only one gene's product. In reality, this is impractical. Polypharmacology is increasingly recognized as common for compounds[12,13], and this is likely to substantially impact the ability of compounds' images to match one of the annotated genes' images. As well, the target annotations of the compounds may not be complete because of undiscovered gene–compound relationships.

We also note limitations in the selection of genes and compounds. This dataset was curated with compounds and genes available in the Broad Institute's Drug Repurposing Hub and in its library of genetic perturbation reagents, respectively, which introduces several biases. First, the set contains only preclinical or clinical compounds with stronger binding and higher specificity than randomly synthesized compounds; however, for the tasks of mechanism of action determination and virtual screening (in which the goal is ultimately to identify such compounds), this is not an overly concerning bias. Second, because of our selection criterion that at least two compounds should target the product of every gene, all compound–gene pairs without a second compound in the Drug Repurposing Hub were excluded, making better-studied compounds more represented. There were also other selection criteria that introduce various biases, such as that the selected compounds should not be a controlled substance. In terms of experimental conditions, we used a single concentration (5 μM) for all compounds, which is not ideal, and we created this dataset in a single experiment at a single facility; this choice minimizes technical variability and therefore maximizes the biological signal in the data, but also limits the potential for generalizability of any models using it as training data. We note that generalizability of models across datasets is often unnecessary in biology experiments, where controls can be included in each experiment; in fact, we recommend that those creating large datasets include the sets of compounds and genes we present here in the experiment to have internal controls and/or landmarks for the assay. Our consortium has adopted this approach in creating our large-scale dataset of 136,000 chemical and genetic perturbations[14].

## Online content

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

## Methods

### Compound and gene selection

The CPJUMP1 dataset consists of images and profiles of cells that were perturbed separately by chemical and genetic perturbations, in which both sets were chosen based on expected matching relationships between them. Chemical perturbations refer to small molecules (compounds) that affect the function of cells, while the genetic perturbations refer either to ORFs that overexpress genes (that is, yield more of the gene's product in the cell) or to guide RNAs that mediate CRISPR-Cas9 (clustered regularly interspaced short palindromic repeats), which cause knockout of gene function (by decreasing production of the gene's product in the cell; the term 'knockout' is used in the field, although in the timescale of these experiments residual protein probably remains, depending on natural rates of protein turnover).

We therefore designed CPJUMP1 such that for each gene, we have one ORF, two CRISPR guides (for all but 15 genes), and one or two compounds that are thought to affect the cell by influencing the function of that gene's product (although they may also influence the function of other genes due to polypharmacology, complicating the signal; Supplementary Tables 8 and 9).

We derived the list of compounds from the Broad Institute's Drug Repurposing Hub dataset[8], a curated and annotated collection of Food and Drug Administration-approved drugs, clinical trial drugs and preclinical tool compounds (Extended Data Fig. 4d). The genes perturbed by genetic perturbations were chosen because their associated proteins are annotated targets of the compounds. The specific criteria for compounds, genetic reagents (considering their on- and off-target effects), and controls are described in the section 'Compound selection criteria', and their layout on the plates is described in the section 'Plate layout design'. After applying the filters and including controls, we selected a total of 303 compounds and 160 genes such that their corresponding perturbations could fit into three 384-well plates with controls.

### Compound selection criteria

We filtered the Drug Repurposing Hub compounds using several criteria, of which three are most important:

- The compounds should target proteins encoded by genes that belong to diverse gene families (Supplementary Table 10). This is because methods for representation learning and gene–compound matching should work well for many different biological pathways, rather than for only a few that are well-characterized and/or easy to predict.
- Each gene product should be targeted by at least two compounds, so that gene–compound matching and compound–compound matching can both be performed using the dataset.
- The requirement that each compound should target only a single gene product, which was considered, but which is difficult to achieve due to polypharmacology (Supplementary Tables 8 and 9), that is, the property for compounds to bind and affect many different gene products in the cell; this is especially common for protein kinase inhibitors in the dataset. Instead, we filtered out only the so-called 'historical compounds' listed in the Chemical Probes Portal[15], which are compounds that are known to be quite non-selective (or not sufficiently potent) compared with other available chemical probes.

Our list of compounds and genes also includes both negative and positive controls. The negative controls for each perturbation modality are, first, the compounds (DMSO, that is, dimethyl sulfoxide, which is the solvent for all of the compounds studied; in other words, all samples will have DMSO added at the same concentration, but the negative controls have no additional compound added); second, ORFs (15 ORFs with the weakest signature in previous image-based profiling experiments[16]; thus, the total number of genes with ORFs is 160 + 15 = 175); and third, CRISPR guides (30 CRISPR guides that target an intergenic site (cutting controls, $n = 3$) or do not have a target sequence that exists in human cells (non-cutting controls, n = 27)).

There are three types of compound positive controls in our list. First, we included chemical probes that are very well-studied and (unlike most compounds) are known to very selectively modulate the target genes' product (poscon_cp)[15]. Second, we included compounds that strongly correlate with the correct genetic perturbation in previous image-based profiling experiments with ORFs[16] and compounds (poscon_orf)[17]. Finally, we included a set of very diverse pairs of compounds with strong intra-pair and weak inter-pair correlations, based on prior experiments (poscon_diverse).

Additionally, compounds were filtered based on availability from at least one of four compound vendors (Sigma, SelleckChem, Tocris and MedChemEx) and genetic reagents via the Broad Insitute's Genetic Perturbation Platform portal. Last, we also excluded compounds on the U.S. Drug Enforcement Agency (DEA) list of controlled substances or the Organisation for the Prohibition of Chemical Weapons (OPCW) list of chemical weapons precursors.

### Target loci selection

We picked the target loci for the CRISPR experiments by selecting the top-two-ranking single-guide RNA sequences that maximize their on-target activity, calculated using the Azimuth 2.0 model[18], and minimize the off-target activity, calculated using the Cutting Frequency Determination score (additional details can be found at https://portals.broadinstitute.org/gpp/public/software/sgrna-scoring-help).

### Compound and gene metadata

A list of CRISPR reagents and their target sequences is available here: https://github.com/jump-cellpainting/2024_Chandrasekaran_NatureMethods/blob/6ba3fcd1495d9e844e4607373a568641981ffcd8/metadata/external_metadata/JUMP-Target-1_crispr_metadata.tsv

A list of ORF reagents and their target sequences is available here: https://github.com/jump-cellpainting/2024_Chandrasekaran_NatureMethods/blob/6ba3fcd1495d9e844e4607373a568641981ffcd8/metadata/external_metadata/JUMP-Target-1_orf_metadata_with_sequence.tsv

A list of compounds with their names, PubChem unique identifier, SMILES (Simplified Molecular Input Line Entry System) string, and gene product targets is available here: https://github.com/jump-cellpainting/2024_Chandrasekaran_NatureMethods/blob/6ba3fcd1495d9e844e4607373a568641981ffcd8/metadata/external_metadata/JUMP-Target-1_compound_metadata_targets.tsv

Infection efficiency data for the ORF and CRISPR regents for each time point and cell type from the CellTiter-Glo cell viability assay are available here: https://github.com/jump-cellpainting/2024_Chandrasekaran_NatureMethods/blob/6ba3fcd1495d9e844e4607373a568641981ffcd8/metadata/external_metadata/CPJUMP1_Infection_Efficiency.xlsx

### Plate layout design

The first plate (Extended Data Fig. 4a) consists entirely of compounds, with two compounds per gene product target. Each compound is in singlicate on the plate except for a dozen or so compounds (poscon_diverse) in duplicate and the negative control DMSO described above, in $n = 64$ replicates. The second plate (Extended Data Fig. 4b) consists entirely of CRISPR reagents, with two guides per gene, each arrayed in its own well and kept separate, with no within-plate replicates; there are two replicates of the 30 CRISPR negative controls described above. The third plate (Extended Data Fig. 4c) consists entirely of ORFs: because there was only one perturbation reagent per gene, there are two replicates of each per plate, plus $n = 4$ replicates of the 15 ORF negative controls. Each plate contains only one type of perturbation modality.

We also considered the impact of edge effects, or plate layout effects, in our design. Edge effects are the technical artifact whereby different samples will yield different behavior depending on where they are located on a plate; generally this is observed mainly in the outer two rows and columns of the plate, and the problem persists despite efforts to mitigate it experimentally[19]. While designing the plate layout, we divided the plate into outer and inner wells, where the outer wells are the two rows and columns closest to the edge of the plate and the inner wells are the rest of the wells on the plate. Then we applied the following constraints to minimize the impact of edge effects: both of the compounds that have the same target will either be in the inner wells or in the outer wells (they will not be split such that one of the compounds is in the inner well while the other is in the outer well); the gene associated with the target of outer well compounds will be in the outer wells of the genetic perturbation plate; and all of the positive control compounds will be in the inner wells. If preferable, with this design, an analysis can be constrained to the inner wells only, to ensure that edge effects have minimal influence on the results.

## Experimental conditions

**Compounds.** The treatment compounds were assayed at 5 µM and the cell seeding density was 1,000 cells per well.

**Open reading frames.** The cell seeding density was 1,625 cells per well, with a media seeding volume of 40 µl per 384 wells. The viral volume was 1 µl virus for 384 wells. The concentration of polybrene was 4 µg ml⁻¹. The media was changed after 24 h, removing polybrene and virus and adding back 50 µl media. No selection was done with blasticidin.

**CRISPR.** The cell seeding density was 350 cells per well, with a media seeding volume of 40 µl for 384 wells. The viral volume was 0.5 µl virus for 384 wells, and the polybrene concentration was 4 µg ml⁻¹. The media was changed after 24 h, removing polybrene and virus and adding back 50 µl media. No selection was done with blasticidin.

## Experimental conditions tested

Although constrained by cost, we captured the compound, ORF and CRISPR plates under several experimental conditions to identify those that improve gene–compound, compound–compound and gene–gene matching (Fig. 3). We did more replicate plates for conditions that were less expensive or that were the most promising, and for the compound and CRISPR plates that had only singlicates of most samples (as compared with ORFs, which had duplicates on the plate). Additionally, we captured plates under many other experimental conditions, listed below, to optimize the experimental conditions. UMAP embeddings of all of the experimental conditions are shown in Extended Data Figs. 5–10.

We note that the cell types commonly used are historical lines derived from two white patients, one male (A549) and one female (U2OS). Therefore, conclusions from these data may hold true only for the demographics or genomics of those persons and not broader groups. They were chosen because the lines are both well-suited for microscopy, and they offer the advantage of enabling direct comparison with extensive prior studies using them.

**Primary group of experimental conditions.** For the primary group of experimental conditions we used four replicate plates of compounds and CRISPR guides and two replicate plates of ORFs (which, as mentioned, contain two replicates in each plate) at two time points and two cell lines each. The short and long time points were different for each perturbation type: for compounds they were 24 h and 48 h; for ORFs they were 48 h and 96 h; and for CRISPR guides they were 96 h and 144 h. The two cell lines were U2OS and A549. For CRISPR experiments, polyclonal A549 and U2OS were used.

**Secondary group of experimental conditions.** The experimental conditions are as follows: we used one A549 96 h ORF plate, in which the cells have been additionally treated with blasticidin (a drug that kills cells that have not been properly infected with the genetic reagent); two replicate plates of the A549 144 h CRISPR plate, in which the cells have been additionally treated with puromycin (a drug that kills cells that have not been properly infected with the genetic reagent); two replicate plates of the A549 48 h compound plate with 20% higher cell seeding density than the baseline; two replicate plates of the A549 48 h compound plate with 20% lower cell seeding density than the baseline; we imaged four replicate plates of the A549 24 h compound plate six additional times to test for photobleaching from repeated imaging; we imaged two replicates of the ORF plates in U2OS and A549 at 96 h and 144 h four additional times, once each on days 1, 4, 14 and 28 after the first imaging, to test the stability of samples over time; and we used four replicate plates of 48 h compound plates in polyclonal A549 cells with Cas9.

## Number of plates and images

Across both of the groups of experimental conditions there are 51 physical plates and 107 plates of images: 40 physical plates and 40 plates of images in the primary group, and 11 physical plates and 67 plates of images in the secondary group. Each plate consists of 384 wells and, on average, nine sites were imaged within each well. At each site, eight (five fluorescent and three brightfield) images were captured. This amounts to nearly 3 million images across the 107 plates.

## Sample preparation and image acquisition

The A549 and U2OS parent lines can be ordered from ATCC directly, but the Cas9 cells are not available due to Broad Institute licensing restrictions. The Cell Painting assay involves staining eight components of cells with six fluorescent dyes that are imaged in five channels: nucleus (Hoechst; DNA), nucleoli and cytoplasmic RNA (SYTO 14; RNA), endoplasmic reticulum (concanavalin A; ER), Golgi and plasma membrane (wheat germ agglutinin (WGA); AGP), mitochondria (MitoTracker; Mito), and the actin cytoskeleton (phalloidin; AGP) (Fig. 1). We optimized the Cell Painting assay described by ref. 6 by changing the concentrations of Hoechst and phalloidin, and combining dye addition and dye permeabilization steps to create Cell Painting v2.5. The changes to the protocol are listed at https://github.com/carpenterlab/2022_Cimini_NatureProtocols/wiki#changes-in-the-official-protocol-to-create-v25-chandrasekaran-et-al-2021. The changes are described in more detail by ref. 7, in which we further developed Cell Painting v3, where we also changed the concentrations of concanavalin A and SYTO14. The images were acquired across five fluorescent channels plus three brightfield planes using a Revvity Opera Phenix HCI microscope in widefield mode with 16 bit depth and with a ×20, 1.0 numerical aperture water immersion lens. Pixel binning was used at 2 × 2, for a final effective pixel size of 0.598 µm.

## Image display

In Fig. 1, each channel was mapped to a final 0–255 look-up table per the following colors and display ranges in Fiji[20]: channel 1 (Mito): Red (1,078–10,191); channel 2 (AGP): Orange Hot (426–22,225); channel 3 (RNA): Yellow (360–33,716); channel 4 (ER): Green (238–14,272); and channel 5 (DNA): Cyan (238–20,508). The final image represents the maximum value in the RGB color space for all five channels, created with Fiji's Composite (Max) mode. No other (linear or nonlinear) adjustments were performed. Panels were assembled using the Magic Montage Fiji tool and channel annotations added in Google Draw.

## Image processing

We used CellProfiler bioimage analysis software (v4.0.6) to process the images using classical algorithms[21]. We corrected for variations in background intensity[22] and then segmented cells, distinguishing between

nuclei and cytoplasm. Then, across the various channels captured, we measured various features of cells across several categories including fluorescence intensity, texture, granularity, density and location (see http://cellprofiler-manual.s3.amazonaws.com/CellProfiler-4.0.6/index.html for more details). Following the image analysis pipeline (see https://github.com/jump-cellpainting/2024_Chandrasekaran_NatureMethods/tree/6ba3fcd1495d9e844e4607373a568641981ffcd8/pipelines for the pipelines), we obtained 5,792 feature measurements from each of more than 75 million cells.

## Image-based profiling

We used cytominer and pycytominer workflows to process the single-cell features extracted using CellProfiler[23–26]. We aggregated the single-cell profiles by computing the median profile, and then normalized the averaged profiles by subtracting the median and dividing by the median absolute deviation (m.a.d.) of each feature. This was done in two ways: using the median and m.a.d. of, first, the negative control wells on the plate (used in the analysis shown here), and second, all the wells on the plate. Finally, we filtered out redundant features (such that no pair of features has Pearson correlation greater than 0.9), as well as features with near-zero variance across all the plates.

## Average precision, mean average precision and fraction retrieved

Average precision serves as our primary metric for both replicability (how distinguishable the replicates of a perturbation are from negative controls) and biological relevance (how distinguishable the sister perturbations are from other perturbations). We measure the similarity between profiles using cosine similarity or absolute value of cosine similarity for cases in which both positive and negative correlations are considered matches. To calculate average precision we first construct a binary ranking of sample and negative control profiles by their cosine similarities. Based on this rank list, calculation of the average precision (AP) score follows the common formulation in the field[27], that is, we average the precision values at each rank $k$ where recall changes:

$$AP = \sum_k (R_k - R_{k-1}) P_k$$

We then assign a $P$ value to each average precision score by a permutation-based significance testing approach. Specifically, we assess the significance of the average precision score against a null distribution built by randomly shuffling the rank list 100,000 times and computing corresponding random average precision scores. These $P$ values are then adjusted for multiple testing using the Benjamini–Hochberg procedure to obtain corrected $P$ values (which we refer to as $q$ values).

We also compute mean average precision by calculating the mean of the average precision scores for each class, where 'class' refers to either a specific perturbation (for replicability) or perturbations associated with the same gene (for biological relevance). We report a per-class mean average precision value along with a combined $q$ value obtained by taking the geometric mean of the $q$ values. Finally, we summarize the mean average precision values for a specific task by calculating the fraction of classes with $q$ values below the significance threshold (0.05), termed 'fraction retrieved'.

## Recommended dataset splits

The methods presented in the benchmarks do not involve any training (we simply use a predetermined similarity metric and hand-engineered features or a pre-trained model) and thus did not require the typical train–validate–test data splits. Depending on the use case, we suggest using different splits. For the two benchmarks, we offer the following guidelines for creating data splits when training is involved.

**Representation learning.** For general representation learning and domain adaptation tasks, one could train on the dataset from one cell line or time point and test it on the other cell line or time point.

**Gene–compound matching.** For gene–compound matching, first, all replicates of a perturbation should be in the same split. Second, for the CRISPR dataset, both guides should be in the same split. Third, three of the compounds (BVT-948, dexamethasone and thiostrepton) have two different identifiers each in the dataset (because of small differences in structures) but the same compound name, therefore each pair should be in the same split. And last, if analyzing data at the single-cell level, all cells from a well should be in the same split.

We provide recommended data splits in https://github.com/jump-cellpainting/2024_Chandrasekaran_NatureMethods/tree/6ba3fcd1495d9e844e4607373a568641981ffcd8/datasplits

## Tools and software used

Data analysis was performed using python code written in a Jupyter notebook environment[28,29]. Python libraries used for data analysis include Numpy, Scipy, scikit-learn and pandas[30–33]. Plots were generated using matplotlib, seaborn and Plotly libraries[34–36]. The Fiji[20] Magic Montage plugin, Lucidchart and Inkscape were used for generating montages, creating schematics and for adding text to images. Two-dimensional representations of image-based profiles in Fig. 2 and Extended Data Figs. 5–10 were generated using UMAP[37].

## Reporting summary

Further information on research design is available in the Nature Portfolio Reporting Summary linked to this article.

## Data availability

Well-level morphological profiles, image analysis pipelines, profile generation pipelines, plate maps and plate and compound metadata, and instructions for retrieving the cell images and single-cell profiles are publicly available online at https://broad.io/cpjump1. The landing page of the GitHub repository for this dataset has relevant additional information: https://broad.io/cpjump1. Cell Painting images and single-cell profiles are available at the Cell Painting Gallery on the Registry of Open Data on AWS (https://registry.opendata.aws/cellpainting-gallery/) under accession number cpg0000-jump-pilot. For well-level aggregated profiles, we use GitHub as the hosting platform and the files are stored in GitLFS. We have released the data with a CC0 license. Source data are provided with this paper.

## Code availability

The code for reproducing the benchmark results, tables and figures are available at https://github.com/jump-cellpainting/2024_Chandrasekaran_NatureMethods/tree/6ba3fcd1495d9e844e4607373a568641981ffcd8/benchmark and https://github.com/jump-cellpainting/2024_Chandrasekaran_NatureMethods/tree/6ba3fcd1495d9e844e4607373a568641981ffcd8/visualization. We have released the code with a BSD 3-Clause license.

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

## Acknowledgements
We thank the more than 100 scientists who have contributed to the organization and scientific direction of the JUMP Cell Painting Consortium. We acknowledge valuable input on data consistency from S. Seal and D. Figueiredo Vidal. We gratefully acknowledge a grant from the Massachusetts Life Sciences Center Bits to Bytes Capital Call program for funding the data production; and funding to support data analysis and interpretation from members of the JUMP Cell Painting Consortium, from the National Institutes of Health (NIH MIRA R35 GM122547 to A.E.C.), and from the Chan Zuckerberg Initiative DAF, an advised fund of the Silicon Valley Community Foundation (2020-225720 to B.A.C.). We also gratefully acknowledge the use of the Revvity Opera Phenix High-Content/High-Throughput imaging system at the Broad Institute, funded by the S10 Grant NIH OD-026839-01.

## Author contributions
S.N.C., B.A.C., A.G., L.M., M.K.-A., J.G.D., B.F., A.S., M.M., D.K., J.B., D.E.R., S.E.S., S.G., S.S. and A.E.C. contributed to the experimental design. A.G., L.M., M.K.-A., B.F., M.M. and D.H. performed the laboratory experiments. S.N.C., B.A.C., N.J., A.A.K., J.A., M.H., H.S.-A. and S.S. performed data and/or image analysis. S.N.C., B.A.C., M.K.-A., S.E.S., S.G., S.S. and A.E.C. contributed to the interpretation of data. S.N.C., J.C.C., A.A.K., S.S. and A.E.C. wrote the paper and all authors contributed to the editing of the paper. B.A.C., S.S. and A.E.C. carried out administration and supervision.

## Competing interests
S.S. and A.E.C. serve as scientific advisors for companies that use image-based profiling and Cell Painting (A.E.C.: Recursion, SyzOnc, Quiver Bioscience; S.S.: Waypoint Bio, Dewpoint Therapeutics, Deepcell) and receive honoraria for occasional talks at pharmaceutical and biotechnology companies. D.K. and S.G. are employees of Merck Healthcare KGaA. All other authors have no competing interests.

## Additional information
**Extended data** are available for this paper at https://doi.org/10.1038/s41592-024-02241-6.

**Correspondence and requests for materials** should be addressed to Shantanu Singh or Anne E. Carpenter.

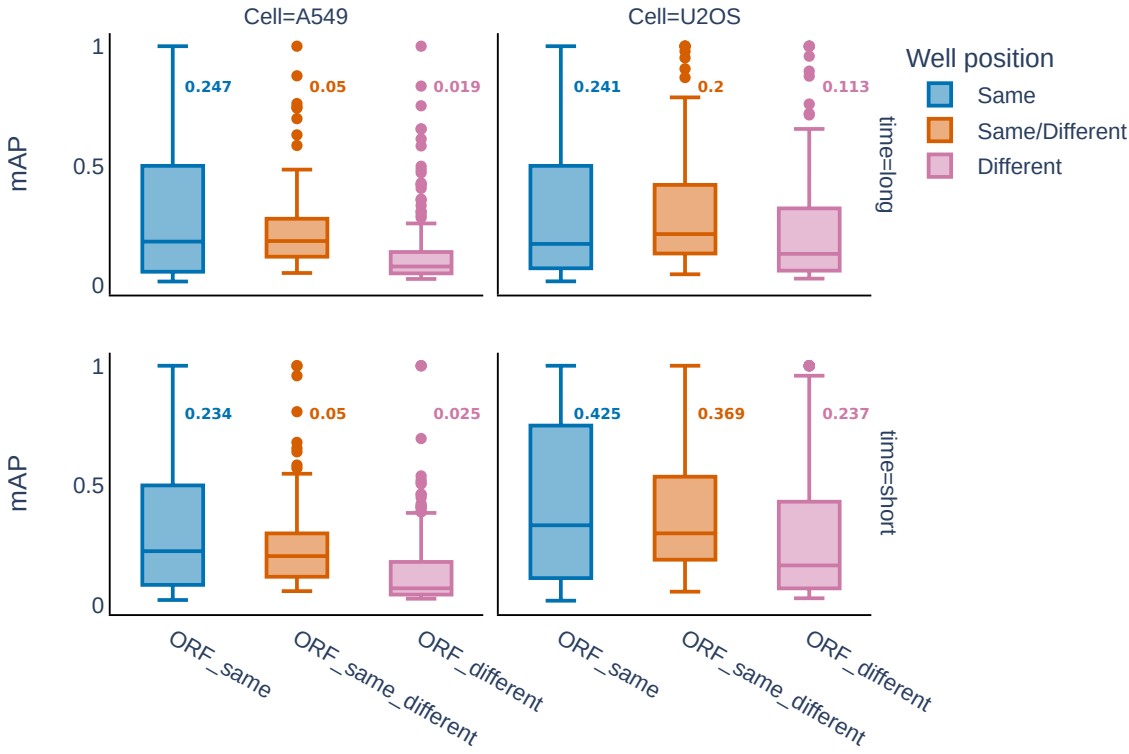

**Extended Data Fig. 1 | Well position effect.** mAP for perturbation detection for ORFs in the same well position (blue), same or different well positions (red) and different well positions (pink); the same or different well positions is what is shown in Fig. 4a in the main text. ORFs in different well positions are affected by plate layout effects, which lowers mAP and FR scores for retrieving replicates against a background of negative control wells. The numerical values shown above each box plot are the fraction of perturbations that can be successfully retrieved (FR) values for each retrieval task. Box plot boundaries are 75th (Q3) and 25th (Q1) percentiles, with whiskers at +/− 1.5 times the interquartile range (Q3–Q1) or the highest or lowest data point. n = 320 biologically independent ORF reagents in the blue boxes and n = 160 biologically independent ORF reagents in the pink and red boxes.

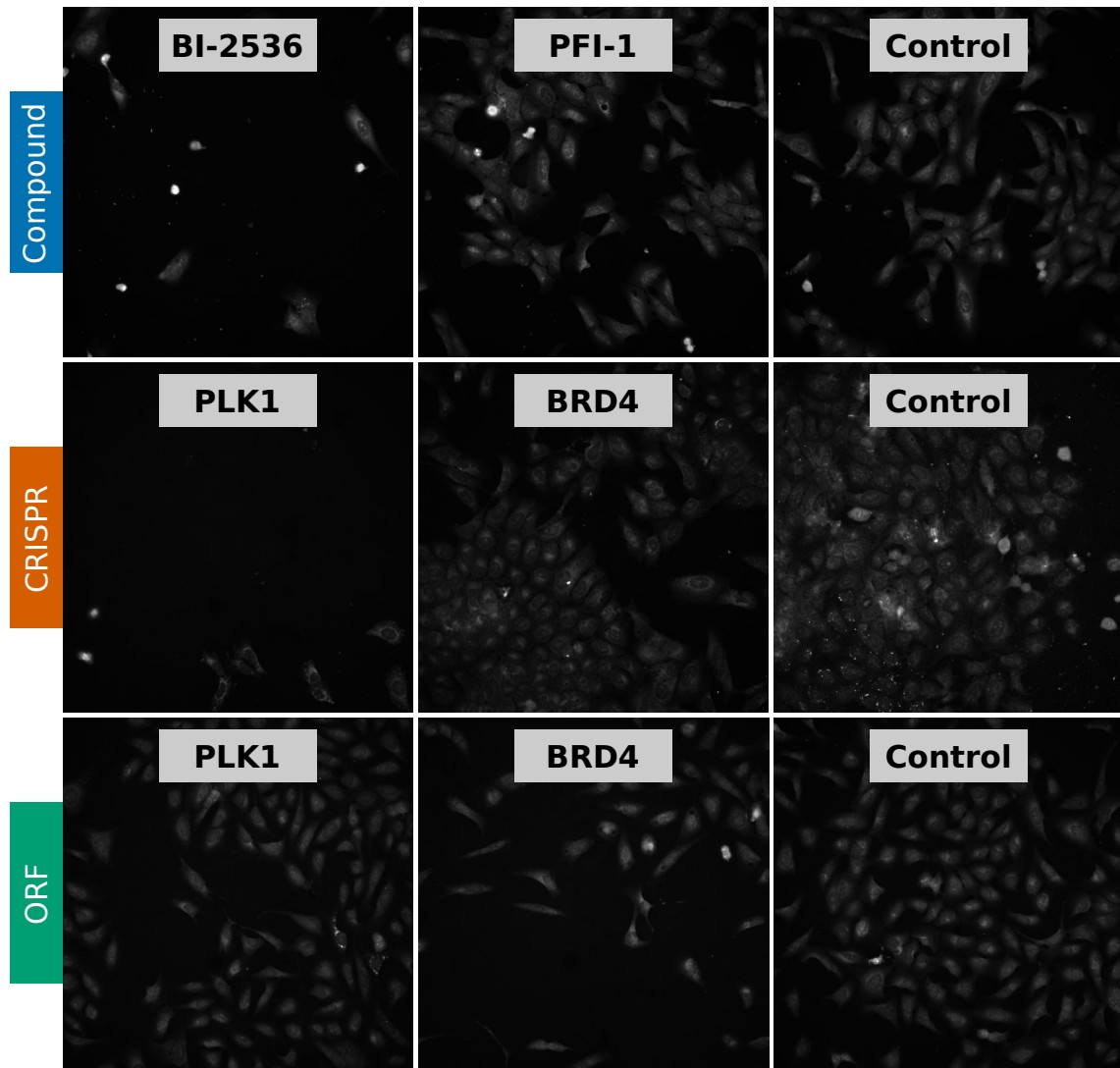

**Extended Data Fig. 2 | Similarity of perturbation impact across modalities for genes and compounds related to the compound BI-2536.** Treatment of U2OS cells with BRD4 inhibitors BI-2536 (multi-target, including PLK1 and BRD4) and PFI-1 (BRD4-specific) is shown (top row; all images are composites of max intensity across all five imaging channels). BI-2536 causes cell death, and this phenotype is mimicked by PLK1 knockout (middle row). In contrast, BRD4 knockout fails to produce a distinct phenotype, death-related or otherwise, as is the case for the BRD4-specific inhibitor PFI-1 (middle column, top and middle row); both profiles are quantitatively similar to negative controls. This implies that BRD4 inhibition has a limited phenotypic impact in this assay under these experimental conditions, allowing the PLK1-inhibiting phenotype of BI-2536 to dominate the profile. BRD4 overexpression, on the other hand, also induces cell death (bottom row) and a profile strongly similar to BI-2536, which could indicate that BRD4 overexpression yields a dominant negative phenotype. Overexpression of PLK1 produces a phenotype that is not distinguishable from negative control. Negative controls for compounds, CRISPR, and ORF reagents are included. These are representative images from one of the four replicate wells of each treatment in the dataset. Wells were sampled from the longer time point for each perturbation modality (Supplementary Table 1).

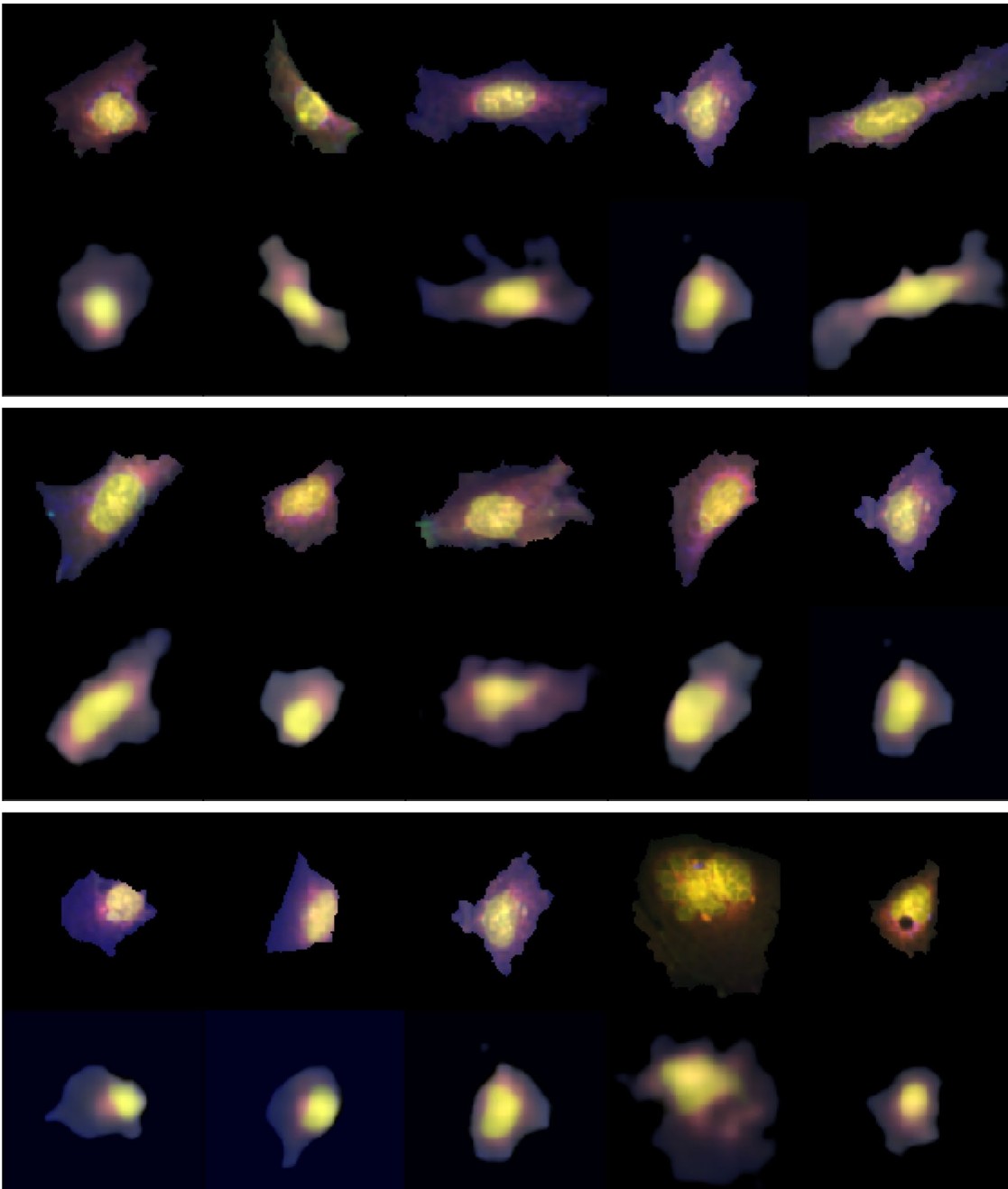

**Extended Data Fig. 3 | CellProfiler features to cell images.** Example single-cell images (first row) and their synthetically generated version (second row) are shown in each sub figure. The synthetic version is generated by each single cell's corresponding CellProfiler measurements. To learn a transformation function from single-cell's CellProfiler extracted features to single-cell images, 7077 single cells were randomly selected from a set of eight diverse compounds (aloxistatin, AMG900, dexamethasone, TC-S-7004, FK-866, LY2109761, NVS-PAK1-1 and quinidine) to train a convolutional neural network (CNN). The set of cell-level Cell Painting measurements was reduced to a non-redundant set of features for five channels of DNA, RNA, ER, AGP and Mito. Location related features and low variance features were excluded. Single-cell images corresponding to each cell's CellProfiler measurements were extracted by image crops of a fixed size (160 pixels) bounding box around the cell's Cells_Location_Center coordinates. CNN model learns the transformation from (3019,1) size CellProfiler features to (128, 128, 5) size images.

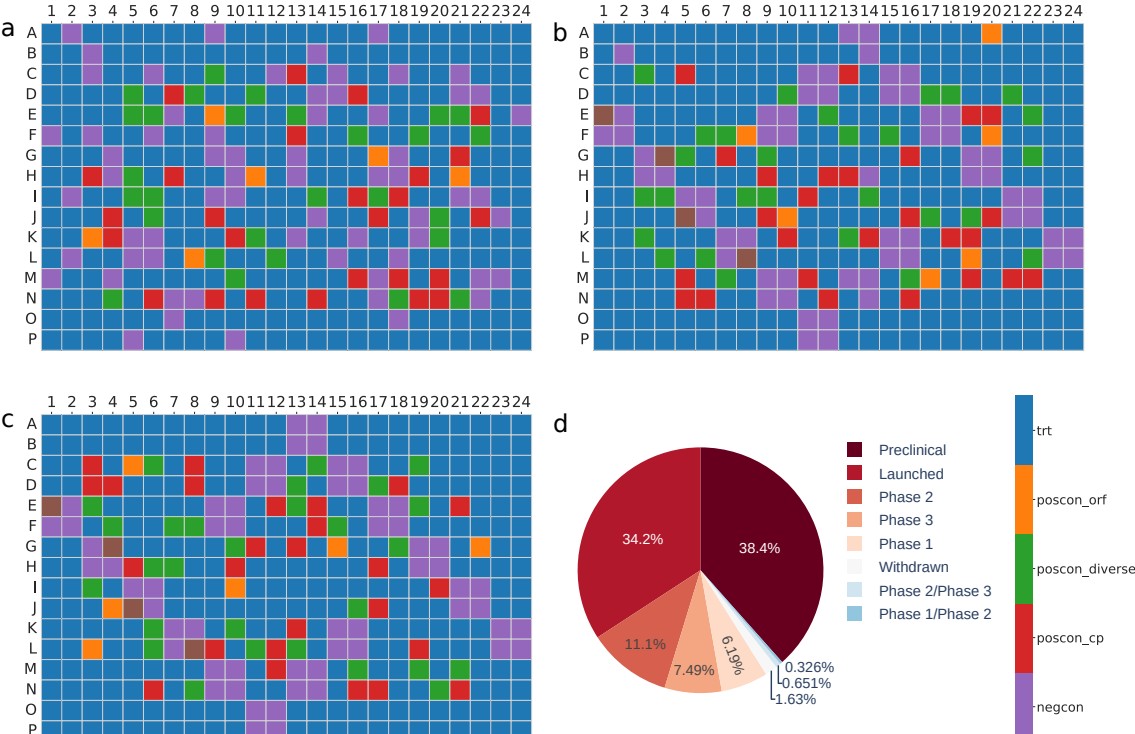

**Extended Data Fig. 4 | Plate maps and overview of compounds' clinical phase status.** Maps in a-c show **a**) Compound plate, **b**) CRISPR plate and **c**) ORF plate. The control wells and the treatment (trt) wells are shown in different colors. Poscon are positive controls (additional details in the Methods section) and negcon is the negative control. **d**) Over a third of the compounds in the dataset have been launched for sale, whereas others have progressed to various stages of human clinical trials.

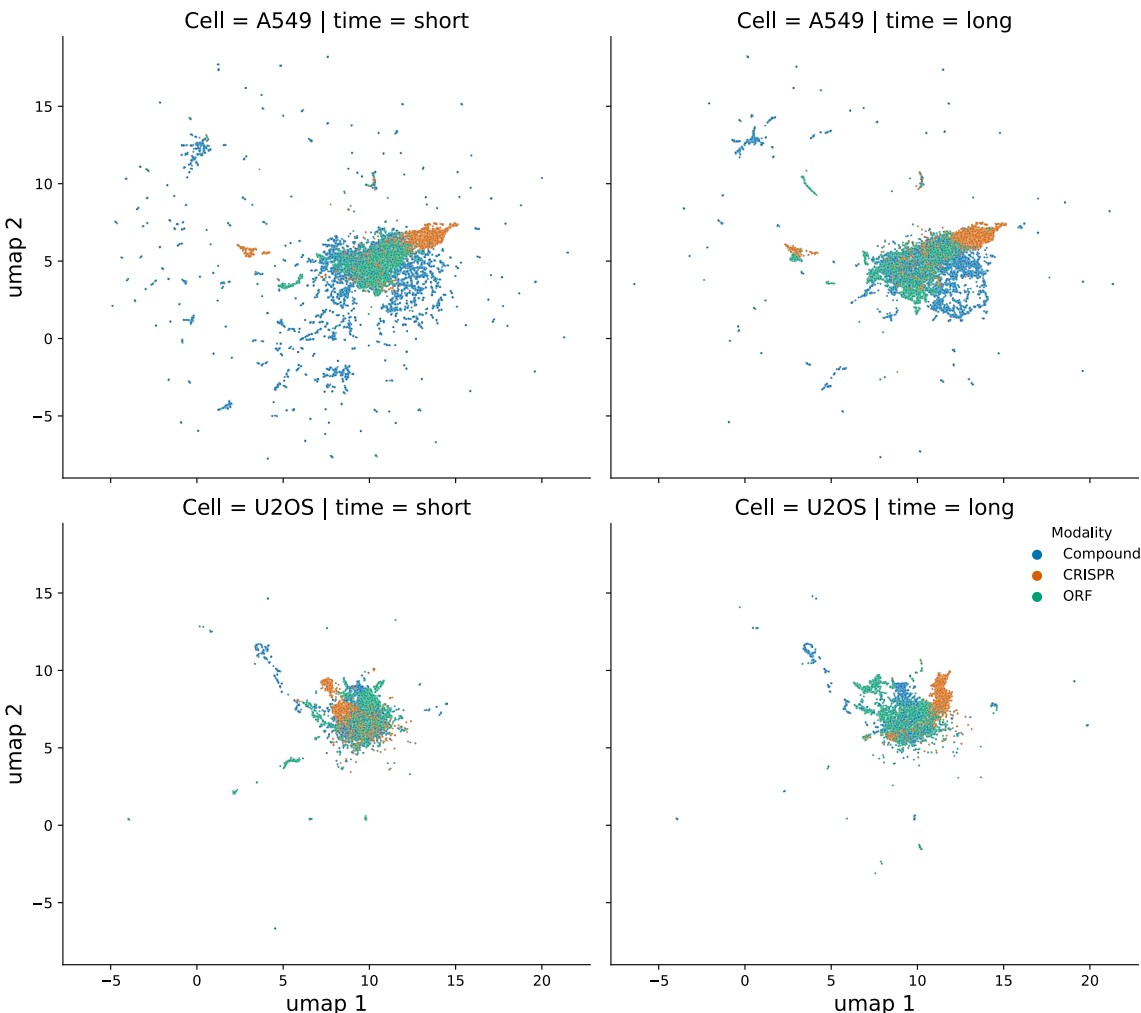

**Extended Data Fig. 5 | Cell type and time, for all tested perturbations and conditions.** The primary group of tested samples in the CPJUMP1 experiment consists of three perturbation modalities (compounds, CRISPR guides and ORFs), two cell types (U2OS and A549) and two time points per perturbation modality (Supplementary Table 1). This UMAP plot includes the CPJUMP1 primary experiment (4 Compound, 4 CRISPR and 2 ORF plates per cell type and time point) plus all other data points from the CPJUMP1 experiments, as outlined in Fig. 3.

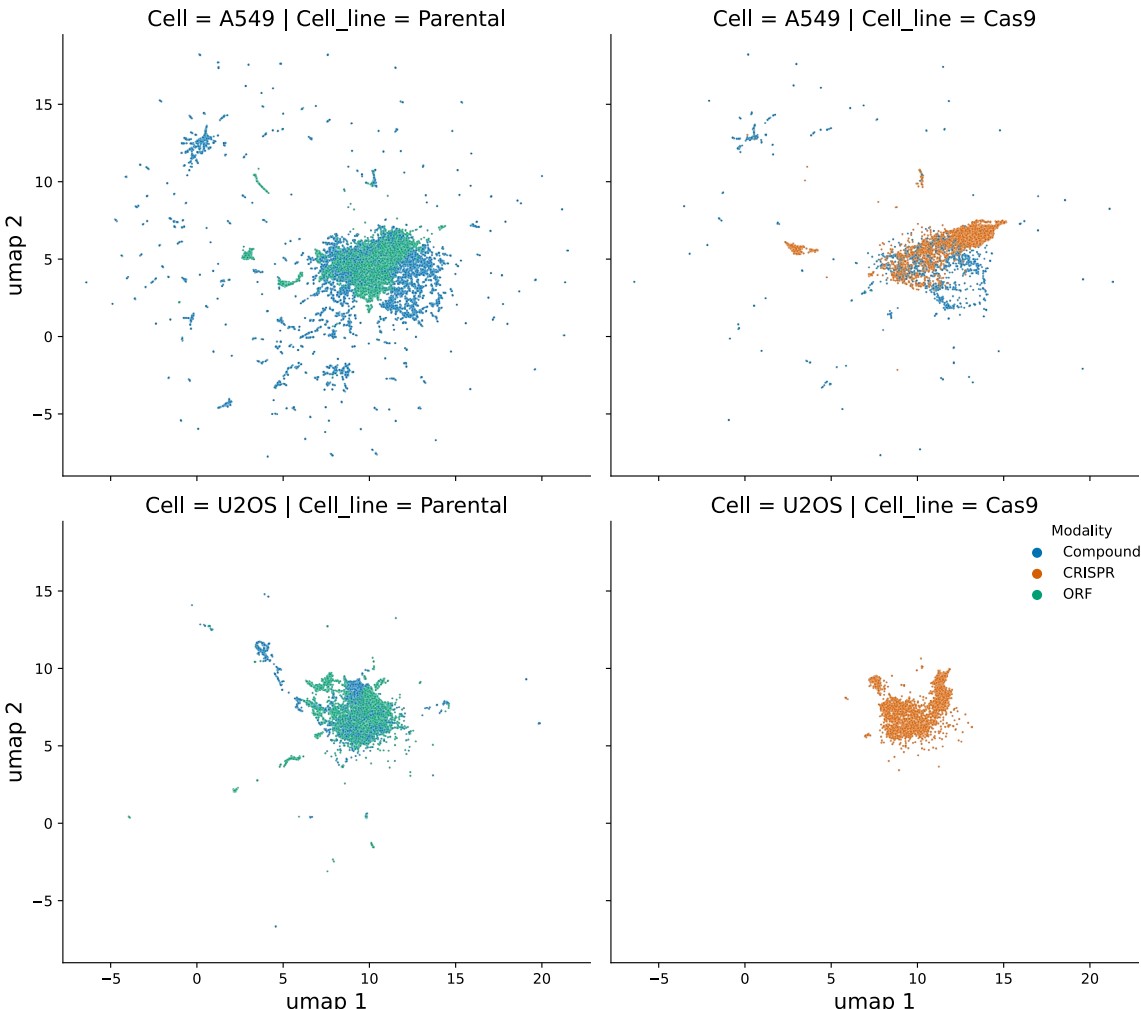

**Extended Data Fig. 6 | Cas9 status.** Parental line is the original cell line with no modifications. Cas9 cell line is a polyclonal cell line expressing Cas9 (used for all CRISPR and one compound experiment). This UMAP plot includes the CPJUMP1 primary experiment (4 Compound, 4 CRISPR and 2 ORF plates per cell type and time point) plus all other data points from the CPJUMP1 experiments, as outlined in Fig. 3.

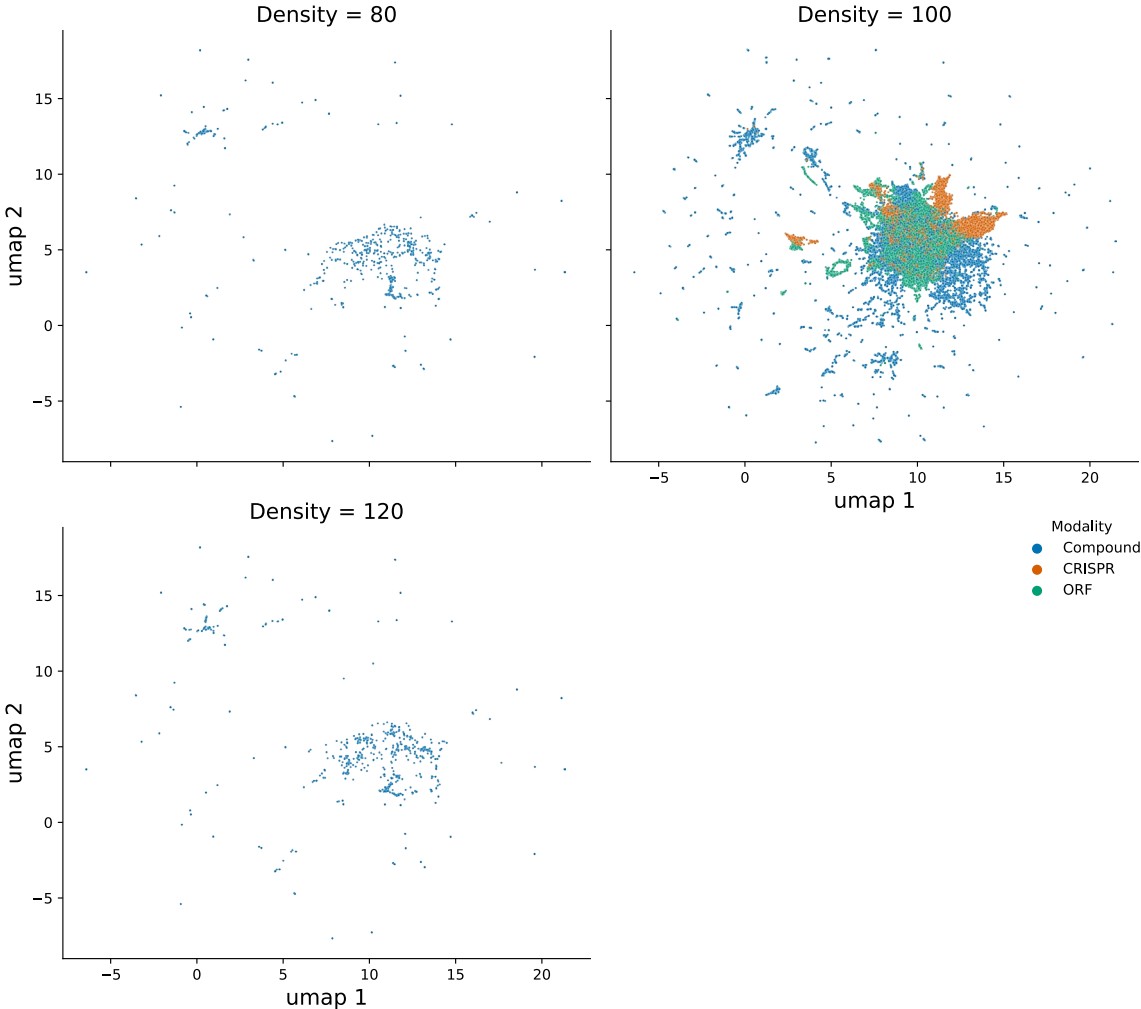

**Extended Data Fig. 7 | Different cell seeding densities.** Experiments were performed with the baseline seeding density (100%; 1000 cells/well), increased seeding density (120%), and decreased seeding density (80%). This UMAP plot includes the CPJUMP1 primary experiment (4 Compound, 4 CRISPR and 2 ORF plates per cell type and time point) plus all other data points from the CPJUMP1 experiments, as outlined in Fig. 3.

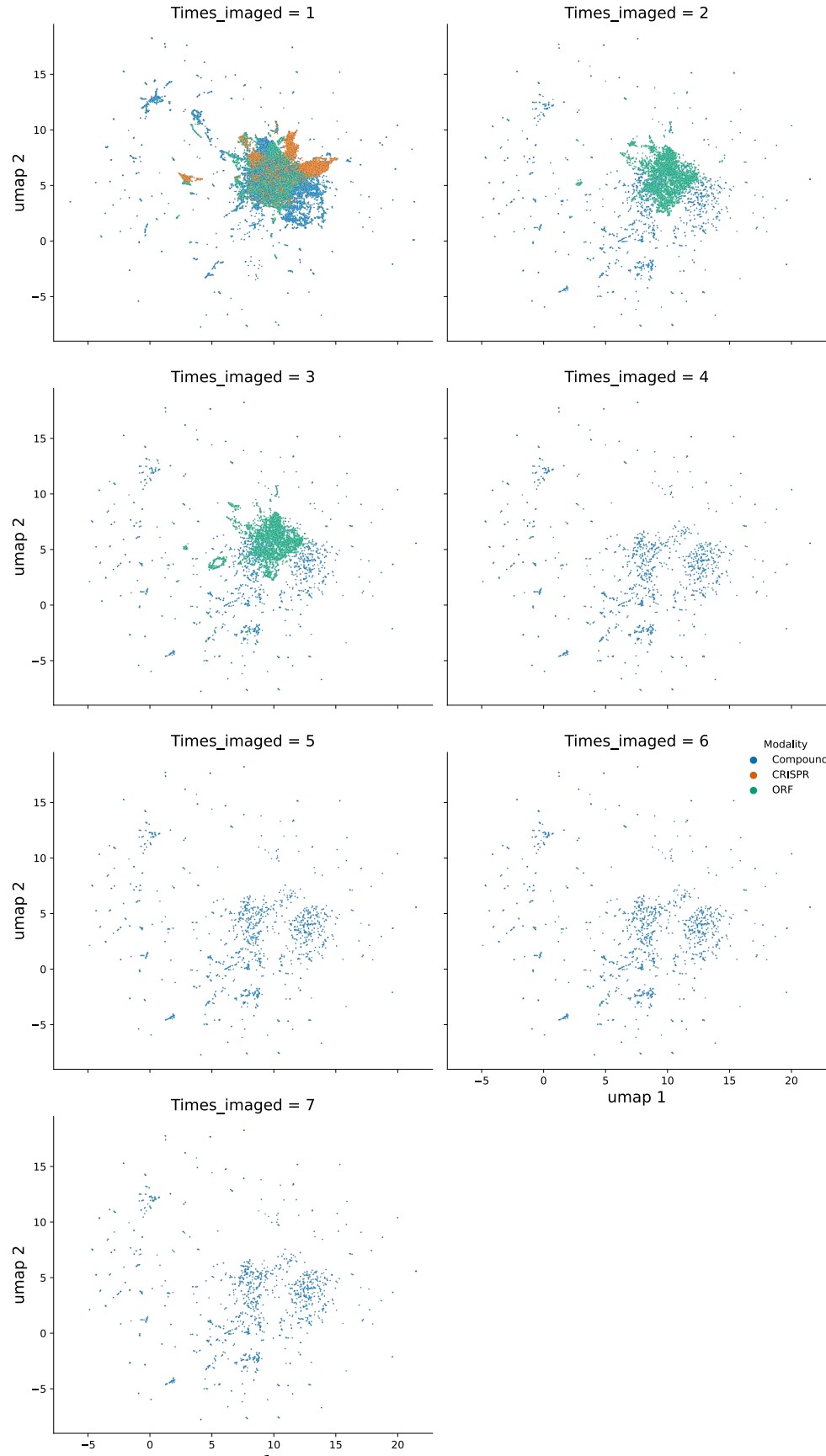

**Extended Data Fig. 8 | Impact of repeat imaging.** Some plates were imaged more than once. This UMAP plot includes the CPJUMP1 primary experiment (4 Compound, 4 CRISPR and 2 ORF plates per cell type and time point) plus all other data points from the CPJUMP1 experiments, as outlined in Fig. 3.

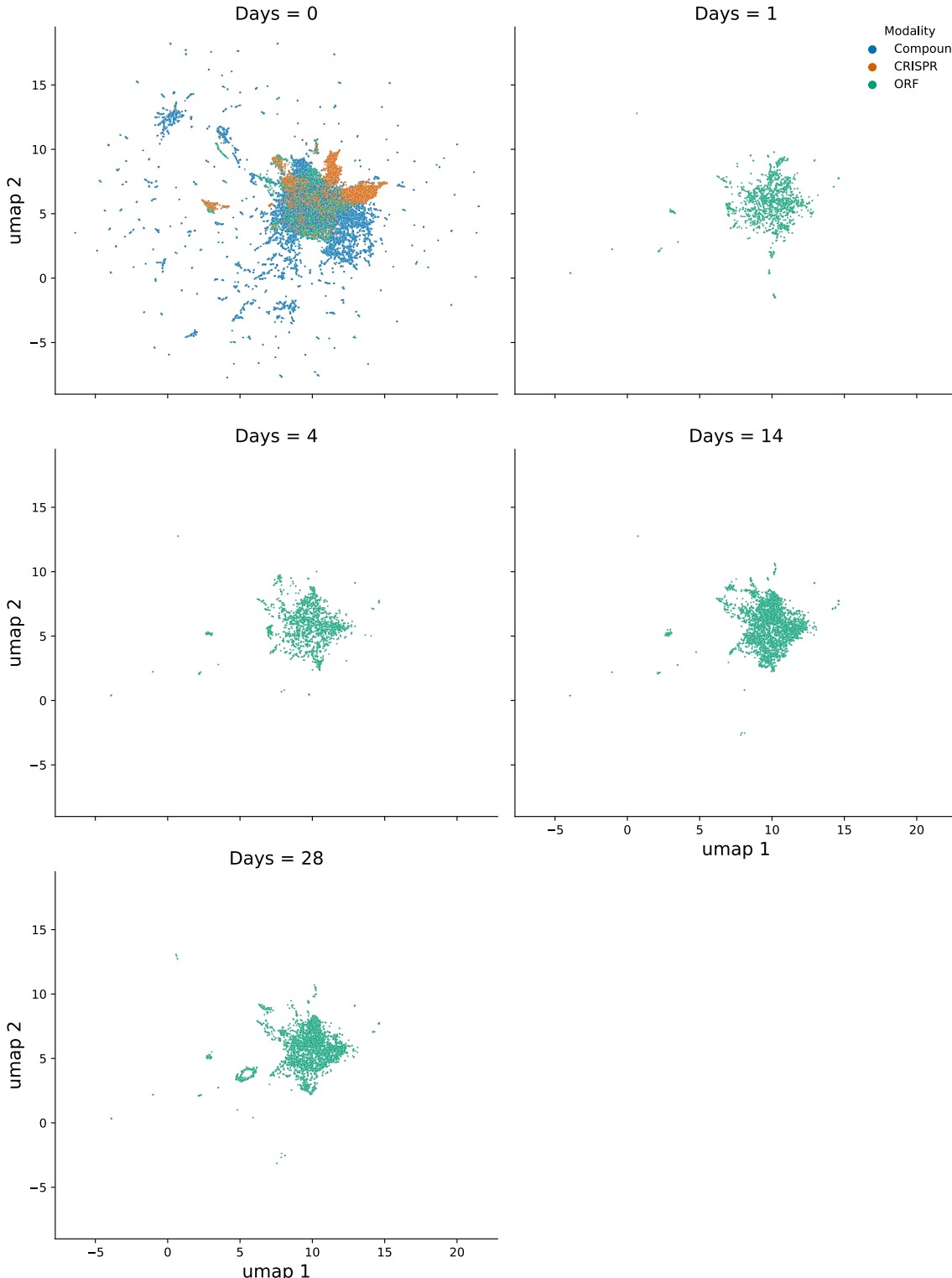

**Extended Data Fig. 9 | Imaging after a time delay.** A subset of plates were imaged after a certain number of days. This UMAP plot includes the CPJUMP1 primary experiment (4 Compound, 4 CRISPR and 2 ORF plates per cell type and time point) plus all other data points from the CPJUMP1 experiments, as outlined in Fig. 3.

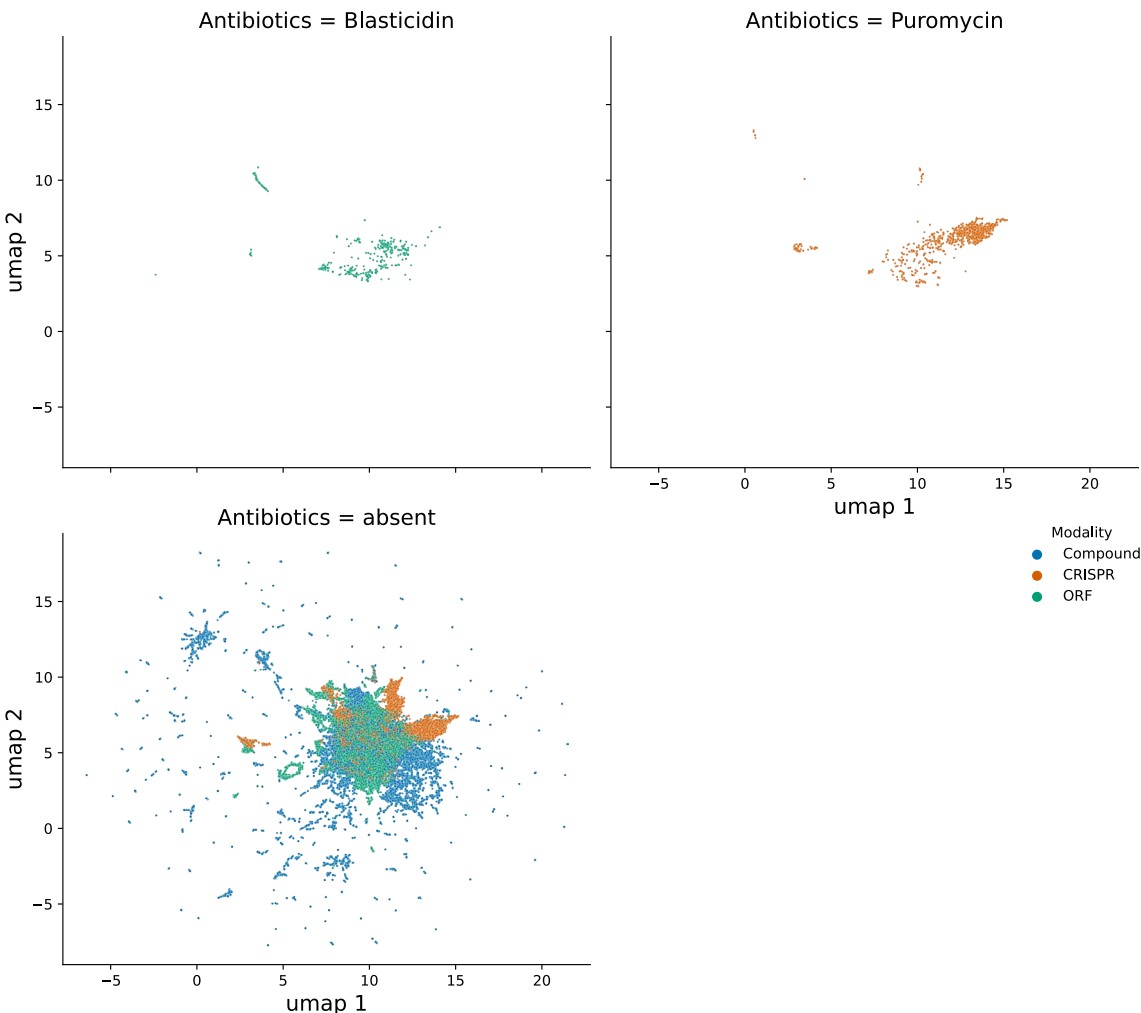

**Extended Data Fig. 10 | Antibiotic selection.** In some CRISPR and ORF plates, cells were selected using antibiotics. This UMAP plot includes the CPJUMP1 primary experiment (4 Compound, 4 CRISPR and 2 ORF plates per cell type and time point) plus all other data points from the CPJUMP1 experiments, as outlined in Fig. 3.

# Reporting Summary

## Statistics

For all statistical analyses, confirm that the following items are present in the figure legend, table legend, main text, or Methods section.

| n/a | Confirmed | |
|---|---|---|
| ☐ | ☒ | The exact sample size (*n*) for each experimental group/condition, given as a discrete number and unit of measurement |
| ☐ | ☒ | A statement on whether measurements were taken from distinct samples or whether the same sample was measured repeatedly |
| ☐ | ☒ | The statistical test(s) used AND whether they are one- or two-sided *Only common tests should be described solely by name; describe more complex techniques in the Methods section.* |
| ☐ | ☒ | A description of all covariates tested |
| ☐ | ☒ | A description of any assumptions or corrections, such as tests of normality and adjustment for multiple comparisons |
| ☐ | ☒ | A full description of the statistical parameters including central tendency (e.g. means) or other basic estimates (e.g. regression coefficient) AND variation (e.g. standard deviation) or associated estimates of uncertainty (e.g. confidence intervals) |
| ☐ | ☒ | For null hypothesis testing, the test statistic (e.g. *F*, *t*, *r*) with confidence intervals, effect sizes, degrees of freedom and *P* value noted *Give P values as exact values whenever suitable.* |
| ☒ | ☐ | For Bayesian analysis, information on the choice of priors and Markov chain Monte Carlo settings |
| ☒ | ☐ | For hierarchical and complex designs, identification of the appropriate level for tests and full reporting of outcomes |
| ☐ | ☒ | Estimates of effect sizes (e.g. Cohen's *d*, Pearson's *r*), indicating how they were calculated |

*Our web collection on statistics for biologists contains articles on many of the points above.*

## Software and code

Policy information about availability of computer code

| Data collection | No software was used to collect data. |
|---|---|
| Data analysis | Data analysis was performed using python code written in a Jupyter notebook environment. Python libraries used for data analysis include Numpy, Scipy, scikit-learn and pandas. Plots were generated using matplotlib, seaborn and Plotly libraries. Fiji7's Magic Montage plugin, Lucidchart and inkscape were used for generating montages, creating schematics and for adding text to images. Two-dimensional representations of image-based profiles in Figs. 2 and Extended Data Figs. 5-10 were generated using UMAP . The code used in this study is available at https://github.com/jump-cellpainting/2024_Chandrasekaran_NatureMethods. It is available for use under the BSD 3-clause license, a permissive open-source license. |

For manuscripts utilizing custom algorithms or software that are central to the research but not yet described in published literature, software must be made available to editors and reviewers. We strongly encourage code deposition in a community repository (e.g. GitHub). See the Nature Portfolio guidelines for submitting code & software for further information.

# Data

Policy information about availability of data

All manuscripts must include a data availability statement. This statement should provide the following information, where applicable:
- Accession codes, unique identifiers, or web links for publicly available datasets
- A description of any restrictions on data availability
- For clinical datasets or third party data, please ensure that the statement adheres to our policy

Well-level morphological profiles, image analysis pipelines, profile generation pipelines, plate maps and plate and compound metadata, and instructions for retrieving the cell images and single cell profiles are publicly available online at https://broad.io/cpjump1.

Cell Painting images and single-cell profiles are available at the Cell Painting Gallery on the Registry of Open Data on AWS (https://registry.opendata.aws/cellpainting-gallery/) under accession number cpg0000-jump-pilot. For well-level aggregated profiles, we use GitHub as the hosting platform and the files are stored in GitLFS.

We have released the data with a CC0 license.

# Human research participants

Policy information about studies involving human research participants and Sex and Gender in Research.

| | |
|---|---|
| Reporting on sex and gender | Not applicable |
| Population characteristics | Not applicable |
| Recruitment | Not applicable |
| Ethics oversight | Not applicable |

Note that full information on the approval of the study protocol must also be provided in the manuscript.

# Field-specific reporting

Please select the one below that is the best fit for your research. If you are not sure, read the appropriate sections before making your selection.

☒ Life sciences ☐ Behavioural & social sciences ☐ Ecological, evolutionary & environmental sciences

For a reference copy of the document with all sections, see nature.com/documents/nr-reporting-summary-flat.pdf

# Life sciences study design

All studies must disclose on these points even when the disclosure is negative.

| | |
|---|---|
| Sample size | Experiments involved cultured cells where the number of cells is governed by the size of the growth vessel; The number of replicates for various biological experiments were chosen based on each expert's knowledge of the variability in the type of experiment. We found from previous experiments that the quality of signal in the data did not improve significantly beyond four replicates. |
| Data exclusions | No data was excluded from the analyses. |
| Replication | All data presented indicates the number of replicates represented. Experiment was performed with four replicates of each treatment for each experimental condition. All replications were successful. |
| Randomization | We designed the experimental plates such that genes that are perturbed by CRISPR and ORF reagents, their products, are always targeted by two compounds. We randomized the location of each sample on the experimental plate as long as they satisfied the following criteria:<br><br>- Both of the compounds that have the same target will either be in the inner wells or in the outer wells. They will not be split such that one of the compounds is in the inner well while the other is in the outer well. where the outer wells are the two rows and columns closest to the edge of the plate and the inner wells are the rest of the wells on the plate.<br>- The gene associated with the target of outer well compounds will be in the outer wells of the genetic perturbation plate.<br>- All the positive control compounds are in the inner wells. |
| Blinding | Experiments involved cells; group allocation is not applicable. |

# Reporting for specific materials, systems and methods

We require information from authors about some types of materials, experimental systems and methods used in many studies. Here, indicate whether each material, system or method listed is relevant to your study. If you are not sure if a list item applies to your research, read the appropriate section before selecting a response.

## Materials & experimental systems

| n/a | Involved in the study |
|-----|------------------------|
| ☒ | ☐ Antibodies |
| ☐ | ☒ Eukaryotic cell lines |
| ☒ | ☐ Palaeontology and archaeology |
| ☒ | ☐ Animals and other organisms |
| ☒ | ☐ Clinical data |
| ☒ | ☐ Dual use research of concern |

## Methods

| n/a | Involved in the study |
|-----|------------------------|
| ☒ | ☐ ChIP-seq |
| ☒ | ☐ Flow cytometry |
| ☒ | ☐ MRI-based neuroimaging |

## Eukaryotic cell lines

Policy information about cell lines and Sex and Gender in Research

| | |
|---|---|
| Cell line source(s) | A549<br>Catalog: CCL-185<br>Lot: 70018877<br><br>U2OS<br>Catalog: HTB-96<br>Lot: 70016635 |
| Authentication | Since the cells were ordered directly from ATCC, we did not authenticate via SNP or STR profiling. |
| Mycoplasma contamination | We tested the parental cells ordered from ATCC for mycoplasma contamination a few months into culturing them to ensure they remained negative. |
| Commonly misidentified lines<br>(See ICLAC register) | To our knowledge neither of these cell lines, U-2 OS nor A549 are commonly misidentified. |

