## [Peer Review File · Nature Methods]

Peer Review Information

Manuscript Title: Three million images and morphological profiles of cells treated with matched chemical and genetic perturbations

Corresponding author name(s): Anne Carpenter, Shantanu Singh

Editorial Notes: None

Reviewer Comments & Decisions:

Decision Letter, initial version:

Dear Anne,

Your Resource, "Three million images and morphological profiles of cells treated with matched chemical and genetic perturbations", has now been seen by three reviewers. As you will see from their comments below, although the reviewers find your work of considerable potential interest, they have raised a number of concerns. We are interested in the possibility of publishing your paper in Nature Methods, but would like to consider your response to these concerns before we reach a final decision on publication. We therefore invite you to revise your manuscript to address these concerns.

We found the reviews on the whole to be constructive. When addressing the comments, we ask that all the relevant sample metadata be provided (drug concentrations, etc) and that you describe how you validated the CRISPR KOs. We think this will help readers assess the quality of the Resource overall. We also ask that you better explain the metrics chosen, and in cases where metrics deviate from field standard definitions, that you clarify and justify any changes (please note, we just published a paper on confusion around AP/mAP in assessing segmentation algorithms <https://www.nature.com/articles/s41592-023-01942-8> that might be of interest to your team).

* include a point-by-point response to the reviewers and to any editorial suggestions

* please underline/highlight any additions to the text or areas with other significant changes to facilitate review of the revised manuscript

- * address the points listed described below to conform to our open science requirements
- * ensure it complies with our general format requirements as set out in our guide to authors at www.nature.com/naturemethods
- * resubmit all the necessary files electronically by using the link below to access your home page

[Redacted]

We hope to receive your revised paper within three months. If you cannot send it within this time, please let us know. In this event, we will still be happy to reconsider your paper at a later date so long as nothing similar has been accepted for publication at Nature Methods or published elsewhere.

OPEN SCIENCE REQUIREMENTS

REPORTING SUMMARY AND EDITORIAL POLICY CHECKLISTS

DATA AVAILABILITY

We strongly encourage you to deposit all new data associated with the paper in a persistent repository where they can be freely and enduringly accessed. We recommend submitting the data to discipline-

specific and community-recognized repositories; a list of repositories is provided here:
<http://www.nature.com/sdata/policies/repositories>

All novel DNA and RNA sequencing data, protein sequences, genetic polymorphisms, linked genotype and phenotype data, gene expression data, macromolecular structures, and proteomics data must be deposited in a publicly accessible database, and accession codes and associated hyperlinks must be provided in the "Data Availability" section.

CODE AVAILABILITY

Please include a "Code Availability" subsection in the Online Methods which details how your custom code is made available. Only in rare cases (where code is not central to the main conclusions of the paper) is the statement "available upon request" allowed (and reasons should be specified).

For more information on our code sharing policy and requirements, please see:
<https://www.nature.com/nature-research/editorial-policies/reporting-standards#availability-of-computer-code>

MATERIALS AVAILABILITY

As a condition of publication in Nature Methods, authors are required to make unique materials

promptly available to others without undue qualifications.

ORCID

Nature Methods is committed to improving transparency in authorship. As part of our efforts in this direction, we are now requesting that all authors identified as 'corresponding author' on published papers create and link their Open Researcher and Contributor Identifier (ORCID) with their account on the Manuscript Tracking System (MTS), prior to acceptance. This applies to primary research papers only. ORCID helps the scientific community achieve unambiguous attribution of all scholarly contributions. You can create and link your ORCID from the home page of the MTS by clicking on 'Modify my Springer Nature account'. For more information please visit please visit www.springernature.com/orcid.

Sincerely,

Rita Strack, Ph.D.
Senior Editor
Nature Methods

Reviewers' Comments:

Reviewer #1:

Remarks to the Author:

The manuscript summarizes the reproducibility and relationship between morphological profiles in the resource dataset CPJUMP1, which includes three million images spanning three types of perturbations on two cell lines. The dataset is expected to be a valuable resource for the research community for further discoveries, such as drug target identification and repositioning.

The manuscript uses mean average precisions (mAP) as a core concept to demonstrate the data's reproducibility and potential. However, the application of this concept may be challenging for non-statisticians. mAP is defined by subtracting the observed average precision (AP) by 95% percentile of AP, obtained from permuted similarities between the query perturbation and the compared perturbations of interest. Additionally, the analysis utilizes the absolute value of cosine similarity between two perturbations, which differs from traditional definitions of mAP used in the AI field. To

enhance comprehensibility for a broader audience, more detailed descriptions are expected in the online method section. For instance, it should clarify whether the rank is based on cosine similarity or absolute cosine similarity.

Despite its value, the reproducibility of the data is questionable. For instance, most mAP values were not statistically significant ($mAP < 0$; Fig 4a) when querying replicates of Open Reading Frame (ORF) perturbations. This warrants further discussion to facilitate the usage of the data.

Another area for improvement in the presentation is the heavy emphasis on artificial intelligence (AI) in the manuscript, while the results only minimally touch upon the topic. Given the weak reproducibility and relationships implied in the results, future AI applications could be very challenging. These concerns could be mitigated by providing more emphasis on some AI examples, such as the one shown in Supplementary Figure 10.

Furthermore, the main manuscript is difficult to follow without sufficient conceptual-level information. Simple references to online methods are not effective in addressing this issue.

Other minor issues:

- 1) It may be worth clarifying that all perturbations were conducted separately, and there were no combinations of perturbations in the experiments. While implied in the text, this point is not clear.
- 2) Clarification is needed regarding the source of three million images since there are only 306 compounds and 160 genes for two cell lines.
- 3) Is the data in the single-cell scale?

Reviewer #2:

Remarks to the Author:

The resource article of Chandrasekaran et al. is an important source for the biological as well as potentially the AI community. The dataset uses the Cell Painting technology to compare chemical with genetic perturbances (CRISPR as well as ORF) in order to identify gene signatures. The dataset is well-annotated, available to the community and can therefore serve as benchmark for further studies. The biological data are carefully controlled, technically sound and image acquisition at different time-points and in different cell lines with appropriate repetitions have been performed. The datasets as well as methods are openly available. Given the current interest in Cell Painting experiments, this is a timely manuscript and an important analysis to be published, although the results are not (yet) always clearly interpretable. The data may however provide the ground stock for the AI community due to their technical robustness.

However, more details of the methods and reagents used should be included in the manuscript. For example, the manuscript should contain details of the critical resources, such as sequences of the Crisper guides and ORFs used as well as names, smiles and source of compounds used in the study. A table with at least the primary pharmacology of the compounds should be provided. While this is an important dataset that should be made available, the current interpretation and comparison of the perturbances is still improvable.

The lack of correlation between the different perturbation methods, in particular between genetic and chemical perturbation is probably surprising. The compound set used contains compounds with varying degree of polypharmacology and although the polypharmacology of the compounds used is mentioned, the complexity of the compounds annotation is 'over'-simplified in the analysis. In fact, a

comparison between the chemical and genetic perturbation may not even be feasible. A compound can for example inhibit the catalytic activity but a genetic perturbation by treating cells with CRISPR guides removes (or reduces) depending on the degree of knockout the whole gene product, also perturbing scaffolding function of the proteins, which will not be mirrored by the inhibitors. As an example, TG-003 not only inhibits CLK1, but also CLK4 as well other CLKs below 1 μM , let alone at 5 μM which was used in the current experiment. The lack of details which compounds were used and assigned to which target makes an evaluation difficult. In addition, no data regarding the degree of CRISPR knockdown was provided. The compound concentration at 5 μM is relatively high. For example, the PLK inhibitor BI-2536 is a pM PLK inhibitor and will show additional inhibition of other kinases at these concentrations. As such the setup of the experiments is somewhat flawed and better annotated inhibitors are available for this kind of analysis. For example, the chemical probes Portal used to weed out the most promiscuous compounds also offers a range of highly selective chemical probes, which moreover could be used together with the matched negative control compounds in addition to the DMSO control.

Minor points:

- Supplementary Figure 10: NVS-PAK1-1 is listed twice
- P.11: The Resource and benchmarks we created with the aim to provide a foundation

Reviewer #3:

Remarks to the Author:

A. Description of the method or tool and any key results

The authors have created the CPJUMP1 dataset comprised of image-based profiling, using the Cell Painting assay, of two cell lines U2OS and A549 following chemical or genetic perturbation (CRISPR knockout or ORF overexpression) at two time points. The chemical perturbations use 306 compounds from the Broad's Drug Repurposing Hub, a curated and annotated collection of FDA-approved drugs, clinical trial drugs, and pre-clinical tool compounds. The genetic perturbations target 160 genes that are known targets of at least one of the 306 compounds. The data and code have been made available and will likely be a valuable resource to the research community, especially those working to develop methods to model data from the Cell Painting assay.

B. Originality and significance of the method and result: if not novel, please include reference

In the Introduction the authors write, "This Resource is unique because there are no other image-based datasets of the Cell Painting assay (described later) that include pairs of genetic and chemical perturbations with their relationships to each other annotated, and that were executed in parallel to minimize technical variations that may confound the signal."

I had trouble following all the qualifiers in this sentence. It sounds like the novelty is coming from this resource being the first data set meeting the following criteria:

1. Uses the Cell Painting assay
2. Includes paired genetic and chemical perturbations
3. The pairs in (2) have their relationships annotated
4. Assays were executed in parallel

Are there previous resources that achieve two or three of these but not all four? A description of previous resources that came the closest and where they fell short would make it clearer what the sets this resource apart.

C. Data & methodology: validity of approach, comparison to available techniques, ease of adoption, quality of data, quality of presentation

The motivation behind the creation of this resource seems to be to attract ML researchers to work more in the field of image-based single-cell profiling, both by making this data set publicly available and highlighting a few simple benchmarking tasks. The latter provides a nice entry point for a ML research with limited background in biology. However, I am concerned that this resource (while large for its field) is not of sufficient size and diversity to be appealing to the broader ML community. To their credit, the authors acknowledge this in the Discussion and Limitations (albeit with a positive spin).

D. Appropriate use of statistics and treatment of uncertainties

The description and use of AP, mAP, and fp were confusing to me. AP is described as a weighted mean; how were the weights calculated? If something non-standard was used, a formula would be helpful. The standardization of the AP values has a permutation testing feel to it; why not use a standard permutation test here (ideally one that accounts for the correlation between features in the data)? Am I understanding correctly that the mAP values are often means over a very small number of AP values? How much agreement do you see between the AP values being averaged? Is zero still the correct reference value for the mAP values when you calculate fp? I think the answer is no, but the difference might be small because you're averaging over only a small number of AP values when calculating the mAP values.

E. Conclusions: robustness, validity, reliability

The main conclusion, that this resource will be useful to the research community, is reasonably well supported by the manuscript.

One moderate concern is how the compound selection criteria might introduce bias into the results of the benchmarking studies when attempting to generalize these results to learn something about cell biology or the biotechnology?

F. Methodological details and algorithms: everything necessary to reproduce the technique?

The authors have done an admirable job of providing scripts to reproduce the results of the paper. The project landing page: <https://broad.io/cpjump1> redirects to https://github.com/jump-cellpainting/2023_Chandrasekaran_submitted

Cell Painting images and single-cell profiles are available at <https://registry.opendata.aws/cellpainting-gallery>

It was not clear to me how the above data are related to the following website: <https://github.com/jump-cellpainting/datasets>

G. Suggested improvements: experiments, comparisons, data for possible revision

As with any Resource paper, it is tempting to suggest many potentially interesting comparisons.

However, in my opinion the authors have struck the correction balance between demonstrating the utility of the Resource without focusing too much attention on specific applications.

H. References: appropriate credit to previous work?

Yes.

I. Clarity and context: lucidity of abstract/summary, appropriateness of abstract, introduction and conclusions

In general the manuscript is well-written and should be understandable by a broad audience. My only suggestion would be to try to reduce the frequency with which the reader is directed to the Online Methods. For example, these two sentences in the Results:

"We used Average Precision (AP) (described further in online methods) to measure how well each sample in the primary group can distinguish its replicates from the negative control samples. We adjusted the AP values for each sample as described in online methods, and then computed the mean Average Precision (mAP) for each perturbation by averaging over its replicates."

Author Rebuttal to Initial comments

Response to comments

Three million images and morphological profiles of cells treated with matched chemical and genetic perturbations

Srinivas Niranj Chandrasekaran, Beth A. Cimini, Amy Goodale, Lisa Miller, Maria Kost-Alimova, Nasim Jamali¹, John G. Doench, Briana Fritchman, Adam Skepner, Michelle Melanson, Alexandr A. Kalinin, John Arevalo, Marzieh Haghighi, Juan Caicedo, Daniel Kuhn, Desiree Hernandez, Jim Berstler, Hamdah Shafqat-Abbasi, David Root, Susanne E. Swalley, Sakshi Garg, Shantanu Singh, Anne E. Carpenter

NMETH-RS52681

We thank the reviewers for their insightful comments and suggestions. In response, we have enhanced the clarity of the metrics described in the manuscript and have provided supplementary metadata, which improved the quality of the manuscript. Additionally, we have revised the formatting of the manuscript to align with the journal's standard formatting guidelines. Below, we have detailed our responses to the reviewers' comments. The reviewer comments are in black and our responses are in blue.

Reviewer 1

The manuscript summarizes the reproducibility and relationship between morphological profiles in the resource dataset CPJUMP1, which includes three million images spanning three types of perturbations on two cell lines. The dataset is expected to be a valuable resource for the research community for further discoveries, such as drug target identification and repositioning.

The manuscript uses mean average precisions (mAP) as a core concept to demonstrate the data's reproducibility and potential. However, the application of this concept may be challenging for non-statisticians. mAP is defined by subtracting the observed average precision (AP) by 95% percentile of AP, obtained from permuted similarities between the query perturbation and the compared perturbations of interest. Additionally, the analysis utilizes the absolute value of cosine similarity between two perturbations, which differs from traditional definitions of mAP used in the AI field. To enhance comprehensibility for a broader audience, more detailed descriptions are expected in the online method section. For instance, it should clarify whether the rank is based on cosine similarity or absolute cosine similarity.

We thank the reviewer for pointing this out. As suggested, we decided to recalculate all the results using a more familiar approach for determining if a computed mAP value is statistically significant by a permutation-based significance testing approach. We compute p values by randomly re-shuffling each rank list 100,000 times and computing the null distribution of AP values. The switch to this approach does not alter the conclusions of the manuscript, but as suggested, is much easier to explain and understand and consistent with recent literature on mAP (Hirling et al. 2023 10.1038/s41592-023-01942-8). We have updated the description of

mAP both in the online methods section and in the results section of the manuscript. We have also updated the manuscript to discuss when cosine similarity and absolute cosine similarity are used for computing mAP (and why).

Despite its value, the reproducibility of the data is questionable. For instance, most mAP values were not statistically significant ($mAP < 0$; Fig 4a) when querying replicates of Open Reading Frame (ORF) perturbations. This warrants further discussion to facilitate the usage of the data.

Another area for improvement in the presentation is the heavy emphasis on artificial intelligence (AI) in the manuscript, while the results only minimally touch upon the topic. Given the weak reproducibility and relationships implied in the results, future AI applications could be very challenging. These concerns could be mitigated by providing more emphasis on some AI examples, such as the one shown in Supplementary Figure 10.

The ability to retrieve replicate samples depends on strong biological signal and weak technical noise. We first want to note that in this experiment, we used a set of genes and compounds that were not filtered for their having any biological signal in this assay. We therefore do not expect 100% of genes to give a reproducible signal; based on past experiments, we expect a minority of genes will yield a detectable phenotype in any one choice of cell line and time point. That said, we believe the low performance of ORFs specifically is due to strong plate layout effects. We have added the following sentences to the results section, acknowledging the retrieval performance, and we suggest an approach for improving retrieval.

“However, we emphasize a strong technical variable that precludes a strong conclusion here: the reduced FR values for ORF may be attributed to plate layout effects, where identical treatments in different rows or columns have dissimilar profiles. This factor amplifies the systematic technical noise in the compound and CRISPR plates due to their particular layout, while it adversely impacts ORFs (online methods).

Retrieving the same position replicates for ORF does (likely artificially) increase FR (Supplementary Figure 1). Plate layout effects can be partly mitigated by mean centering every feature at each well position, though we have not applied the correction to this dataset.”

We also note that a new, more sophisticated model for matching perturbation profiles yields significant improvement over the cosine similarity-based retrieval used in our study. We have added the following sentence to the discussion section to convey this:

“Even with the few hundred perturbations in this dataset, our recent research finds that a transformer-based classification model outperforms the baseline benchmark in this study (unpublished data).”

We’ve also acknowledged the limited examples of machine learning in this paper:

“While our results may only touch upon the potential applications of machine learning, our emphasis is a strategic appeal to the ML community.”

Furthermore, the main manuscript is difficult to follow without sufficient conceptual-level information. Simple references to online methods are not effective in addressing this issue.

We have made changes throughout the manuscript to provide additional details and to improve its readability.

It may be worth clarifying that all perturbations were conducted separately, and there were no combinations of perturbations in the experiments. While implied in the text, this point is not clear.

We have added text, where appropriate, to convey more explicitly that cells were treated with each perturbation modality separately.

Clarification is needed regarding the source of three million images since there are only 306 compounds and 160 genes for two cell lines.

We have added a new section, "Number of plates and images", to the online methods with this information. We have also added additional information to the text describing Figure 3 to clarify this. Briefly, there are 8 images (5 fluorescent + 3 brightfield) at each of the 9 sites in each well of a 384 well plate. There are 51 plates and an additional 56 plates of images (a subset of the 51 plates that were imaged multiple times to evaluate different imaging conditions).

Please note: We resolved an open issue about the number of compounds in the dataset. We have updated this to n=303. Briefly: in three cases, we had two occurrences of what are essentially the same compound. Please see <https://github.com/jump-cellpainting/JUMP-Target/issues/9#issuecomment-1787512329> for further details.

Is the data in the single-cell scale?

The image-based profiles are available at both the single cell level and well-level (median aggregated, there are ~100-200 cells per image). We have updated the introduction section with the following sentence:

"Here, we describe our design and creation of a benchmark dataset via a single large experiment, CPJUMP1. The dataset comprises roughly three million images of cells, image-based profiles of seventy five million single cells, and well-level aggregated profiles"

Reviewer 2

The resource article of Chandrasekaran et al. is an important source for the biological as well as potentially the AI community. The dataset uses the Cell Painting technology to compare chemical with genetic perturbances (CRISPR as well as ORF) in order to identify gene

signatures. The dataset is well-annotated, available to the community and can therefore serve as benchmark for further studies. The biological data are carefully controlled, technically sound and image acquisition at different time-points and in different cell lines with appropriate repetitions have been performed. The datasets as well as methods are openly available. Given the current interest in Cell Painting experiments, this is a timely manuscript and an important analysis to be published, although the results are not (yet) always clearly interpretable. The data may however provide the ground stock for the AI community due to their technical robustness.

However, more details of the methods and reagents used should be included in the manuscript. For example, the manuscript should contain details of the critical resources, such as sequences of the Crispr guides and ORFs used as well as names, smiles and source of compounds used in the study. A table with at least the primary pharmacology of the compounds should be provided. While this is an important dataset that should be made available, the current interpretation and comparison of the perturbances is still improvable. The lack of details which compounds were used and assigned to which target makes an evaluation difficult.

We thank the reviewer for bringing it to our attention that this valuable metadata was not easily findable. We created the following section in the online methods of the manuscript to provide links to all the useful metadata (the repository name will be updated concurrent with the manuscript if the paper is accepted):

“Compound and gene metadata

A list of CRISPR reagents and their target sequences is available here:

https://github.com/jump-cellpainting/2023_Chandrasekaran_submitted/blob/5c6fcf9dc70a85176f5afc5263acbc230d90ca40/metadata/external_metadata/JUMP-Target-1_crispr_metadata.tsv

A list of ORF reagents and their target sequences is available here:

https://github.com/jump-cellpainting/2023_Chandrasekaran_submitted/blob/5c6fcf9dc70a85176f5afc5263acbc230d90ca40/metadata/external_metadata/JUMP-Target-1_orf_metadata_with_sequence.tsv

A list of compounds with their names, PubChem ID, SMILES and gene targets is available here:

https://github.com/jump-cellpainting/2023_Chandrasekaran_submitted/blob/5c6fcf9dc70a85176f5afc5263acbc230d90ca40/metadata/external_metadata/JUMP-Target-1_compound_metadata_targets.tsv

Infection efficiency data for the ORF and CRISPR reagents for each time point and cell type from CellTiter-Glo cell viability assay is available here: https://github.com/jump-cellpainting/2023_Chandrasekaran_submitted/blob/5c6fcf9dc70a85176f5afc5263acbc230d90ca40/metadata/external_metadata/CPJUMP1_Infection_Efficiency.xlsx”

The lack of correlation between the different perturbation methods, in particular between genetic and chemical perturbation is probably surprising. The compound set used contains compounds with varying degree of polypharmacology and although the polypharmacology of the compounds used is mentioned, the complexity of the compounds annotation is 'over'-simplified in the analysis. In fact, a comparison between the chemical and genetic perturbation may not even be feasible. A compound can for example inhibit the catalytic activity but a genetic perturbation by treating cells with CRISPR guides removes (or reduces) depending on the degree of knockout the whole gene product, also perturbing scaffolding function of the proteins, which will not be mirrored by the inhibitors. As an example, TG-003 not only inhibits CLK1, but also CLK4 as well other CLKs below 1 μM , let alone at 5 μM which was used in the current experiment.

In addition, no data regarding the degree of CRISPR knockdown was provided. The compound concentration at 5 μM is relatively high. For example, the PLK inhibitor BI-2536 is a pM PLK inhibitor and will show additional inhibition of other kinases at these concentrations. As such the setup of the experiments is somewhat flawed and better annotated inhibitors are available for this kind of analysis. For example, the chemical probes Portal used to weed out the most promiscuous compounds also offers a range of highly selective chemical probes, which moreover could be used together with the matched negative control compounds in addition to the DMSO control.

The reviewer accurately captures many of the reasons why compounds may not match a gene annotated as its target; this limitation has previously been noted in the literature (Rohban et al. 2022 10.1016/j.cels.2022.08.003). Nevertheless, even though the gene-compound connections cannot serve as literal ground "truth", this matching problem is still a useful choice for evaluation – in fact, it is one of the few scenarios for assessing whether the method correctly matches samples that might have similar biological impact (for genes, one can use functional annotations to match to other genes, and likewise compounds to compounds). We note that in our unpublished recent work, more complex matching approaches are more successful; nevertheless, we augmented our originally brief section on the fact that gene target-compound matching is not expected to work most of the time, and why it is still useful as an evaluation task and as an application in drug discovery (where high failure rates for various steps are not unusual!)

Still, we are hopeful that even though there does not exist a ground truth set of gene-compound pairs that ought to match perfectly (though Chemical Probes Portal comes close), this dataset can nevertheless be used to improve methods to detect those relationships that *can* be modeled in this way. Concerning the use of Chemical Probes portal, we had to use compounds available in the drug repurposing hub Broad library and genetic reagents likewise available in the Broad library for practical reasons (cost and time) and were unable to procure all the Chemical Probes portal pairs (including passing our other filters, such as having two compounds per gene target). We found that there were only 36 genes, many from the same gene family, that were known to be targeted by chemical probes with high confidence in the Chemical Probes Portal. Of these, we included 13 genes in our experiment. Regarding the use of a single concentration (5 μM), practically speaking, we needed to choose a single concentration for all compounds, which is

non-ideal as stated, which we now acknowledge in the discussion section. Discussion among the 10 pharmaceutical companies in the Consortium led to the decision to use 5 uM (after much debate!). We recognize the tradeoff of specificity for some compounds. We also do not know the degree of CRISPR knockdown (We may have to ask GPP if this information is available), nor ORF overexpression for each reagent within this experiment because it is extremely expensive to measure protein levels nor even mRNA levels across a large library (per the Broad Genetic Perturbation Platform). But, we have now provided a link to the infection efficiency data for both ORF and CRISPR reagents in the manuscript (https://github.com/jump-cellpainting/2023_Chandrasekaran_submitted/blob/5c6fcf9dc70a85176f5afc5263acbc230d90ca40/metadata/external_metadata/CPJUMP1_Infection_Efficiency.xlsx). Some CRISPR and ORF reagents have low infection efficiencies; filtering them out may improve signal.

Supplementary Figure 10: NVS-PAK1-1 is listed twice

We have fixed this typo.

P.11: The Resource and benchmarks we created with the aim to provide a foundation

We have revised the text to convey the intended meaning.

Reviewer 3

The authors have created the CPJUMP1 dataset comprised of image-based profiling, using the Cell Painting assay, of two cell lines U2OS and A549 following chemical or genetic perturbation (CRISPR knockout or ORF overexpression) at two time points. The chemical perturbations use 306 compounds from the Broad's Drug Repurposing Hub, a curated and annotated collection of FDA-approved drugs, clinical trial drugs, and pre-clinical tool compounds. The genetic perturbations target 160 genes that are known targets of at least one of the 306 compounds. The data and code have been made available and will likely be a valuable resource to the research community, especially those working to develop methods to model data from the Cell Painting assay.

In the Introduction the authors write, "This Resource is unique because there are no other image-based datasets of the Cell Painting assay (described later) that include pairs of genetic and chemical perturbations with their relationships to each other annotated, and that were executed in parallel to minimize technical variations that may confound the signal." I had trouble following all the qualifiers in this sentence. It sounds like the novelty is coming from this resource being the first data set meeting the following criteria:

- 1. Uses the Cell Painting assay*
- 2. Includes paired genetic and chemical perturbations*
- 3. The pairs in (2) have their relationships annotated*
- 4. Assays were executed in parallel*

Are there previous resources that achieve two or three of these but not all four? A description of previous resources that came the closest and where they fell short would make it clearer what the sets this resource apart.

We simplified the wording as follows:

"This Resource is unique because there are no other image-based datasets of the Cell Painting assay (described later) that include pairs of annotated genetic and chemical perturbations performed side by side, under different experimental conditions such as different cell types, time points, and imaging conditions. These were also executed in parallel to minimize technical variations that may confound the signal. There are many public Cell Painting datasets (for example, <https://github.com/broadinstitute/cellpainting-gallery>) but we are only aware of one with genetic and chemical perturbation types run in parallel (RxRx3, <https://www.rxx.ai/rxx3>), it has not been provided with gene-compound relationship annotations; any that exist would be scattered across many plates and batches. As well, it includes only a single cell type, time point, and imaging condition."

The motivation behind the creation of this resource seems to be to attract ML researchers to work more in the field of image-based single-cell profiling, both by making this data set publicly available and highlighting a few simple benchmarking tasks. The latter provides a nice entry point for a ML research with limited background in biology. However, I am concerned that this resource (while large for its field) is not of sufficient size and diversity to be appealing to the broader ML community. To their credit, the authors acknowledge this in the Discussion and Limitations (albeit with a positive spin).

This is a fair critique and pointed us to adjust the main text to emphasize an aspect of the dataset more: the focus of the analyses in this manuscript was retrieval performance of the perturbations (303 chemical compounds and 160 ORF and CRISPR reagents). But, there are other experimental conditions such as multiple cell lines, time points and photo bleaching that can be useful for developing style transfer methods. Furthermore, Moshkov et al. (<https://doi.org/10.1101/2022.08.12.503783>) show that even with 409 compounds and 79 genetic perturbation reagents that have a strong signature, they were able to train a CNN that outperformed CellProfiler derived image-based features. Hence, we believe that this dataset can enable learning useful representations.

The description and use of AP, mAP, and fp were confusing to me. AP is described as a weighted mean; how were the weights calculated? If something non-standard was used, a formula would be helpful. The standardization of the AP values has a permutation testing feel to it; why not use a standard permutation test here (ideally one that accounts for the correlation between features in the data)? Am I understanding correctly that the mAP values are often means over a very small number of AP values? How much agreement do you see between the AP values being averaged? Is zero still the correct reference value for the mAP values when you calculate fp? I think the answer is no, but the difference might be small because you're averaging over only a small number of AP values when calculating the mAP values.

To avoid confusing the reader, we have now shifted to a more familiar approach for determining if a computed mAP value is statistically significant by permutation testing. Specifically, we compute p values by randomly re-shuffling each rank list 100,000 times and computing the null distribution of AP values. The switch to the new metric does not alter the results or the conclusions of the manuscript. We have updated the description of mAP both in the online methods section, where we corrected the definition of AP and included the formula, and updated the results section of the manuscript correspondingly. Fraction Retrieved (FR), which we used instead of fp, is the fraction of labels that can be correctly retrieved against the reference, which is now determined by whether the FDR corrected p value for a label is below the significance threshold (0.05).

We have also clarified that we use the standard formulation of Average Precision as recommended by <https://www.nature.com/articles/s41592-023-01942-8>.

The main conclusion, that this resource will be useful to the research community, is reasonably well supported by the manuscript.

One moderate concern is how the compound selection criteria might introduce bias into the results of the benchmarking studies when attempting to generalize these results to learn something about cell biology or the biotechnology?

We have added the following text to the discussion to acknowledge the bias that our compound/perturbation selection strategy may have introduced.

"This dataset was curated with compounds and genes available in the Broad's drug repurposing hub and its library of genetic perturbation reagents, respectively, which introduces several biases. First, the set contains only preclinical/clinical compounds with stronger binding and higher specificity than randomly synthesized compounds; however, for the tasks of mechanism of action determination and virtual screening (where the goal is ultimately to identify such compounds), this is not an overly concerning bias. Second, because of our selection criterion that at least two compounds should target every gene, all compound-gene pairs without a second compound in the repurposing hub were excluded, making better-studied compounds more represented. There were also other selection criteria that introduce various biases, such as, the selected compounds should not be a controlled substance."

The authors have done an admirable job of providing scripts to reproduce the results of the paper. The project landing page: <https://broad.io/cpjump1> redirects to https://github.com/jump-cellpainting/2023_Chandrasekaran_submitted

Cell Painting images and single-cell profiles are available at <https://registry.opendata.aws/cellpainting-gallery>

It was not clear to me how the above data are related to the following website: <https://github.com/jump-cellpainting/datasets>

The CPJUMP1 dataset in the current manuscript is a subset of the JUMP dataset described on that website. To avoid confusion, we have removed the link from the manuscript.

As with any Resource paper, it is tempting to suggest many potentially interesting comparisons. However, in my opinion the authors have struck the correction balance between demonstrating the utility of the Resource without focusing too much attention on specific applications.

In general the manuscript is well-written and should be understandable by a broad audience. My only suggestion would be to try to reduce the frequency with which the reader is directed to the Online Methods. For example, these two sentences in the Results:

"We used Average Precision (AP) (described further in online methods) to measure how well each sample in the primary group can distinguish its replicates from the negative control samples. We adjusted the AP values for each sample as described in online methods, and then computed the mean Average Precision (mAP) for each perturbation by averaging over its replicates."

We thank the reviewer for this suggestion. We have added more text to the main text of the manuscript, where appropriate, to avoid frequently directing the reader to the online methods.

Decision Letter, first revision:

Dear Anne,

Thank you for submitting your revised manuscript "Three million images and morphological profiles of cells treated with matched chemical and genetic perturbations" (NMETH-RS52681A). It has now been seen by the original referees and their comments are below. The reviewers find that the paper has improved in revision, and therefore we'll be happy in principle to publish it in Nature Methods, pending minor revisions to satisfy the referees' final requests and to comply with our editorial and formatting guidelines.

TRANSPARENT PEER REVIEW

Please note: we allow redactions to authors' rebuttal and reviewer comments in the interest of confidentiality. If you are concerned about the release of confidential data, please let us know specifically what information you would like to have removed. Please note that we cannot incorporate redactions for any other reasons. Reviewer names will be published in the peer review files if the reviewer signed the comments to authors, or if reviewers explicitly agree to release their name. For more information, please refer to our FAQ page.

ORCID

Sincerely,
Rita

Rita Strack, Ph.D.
Senior Editor

Nature Methods

Reviewer #1 (Remarks to the Author):

The concerns raised in my previous review have been adequately addressed.

Reviewer #2 (Remarks to the Author):

The authors have addressed most of this reviewers concerns and updated the manuscript accordingly. There are only 2 points remaining:

- I would like to ask the authors to update throughout the manuscript any wording referring to a 'compound targeting a gene' as these small molecules target proteins.
- It is surprising that BI-2536 is the top positively correlated gene-compound match as the inhibitor at 5 μ M also inhibits BRD4, inhibition of which produces a very strong phenotype, a point worth discussing.

Reviewer #3 (Remarks to the Author):

The authors have addressed all of my comments and concerns.

Author Rebuttal, first revision:

We thank the reviewers for their suggestions. We have incorporated the suggested changes to the manuscript and have also responded to their concerns below. The reviewer comments are in black/bold and our responses are in blue.

The authors have addressed most of this reviewers concerns and updated the manuscript accordingly.

There are only 2 points remaining:

I would like to ask the authors to update throughout the manuscript any wording referring to a 'compound targeting a gene' as these small molecules target proteins.

Good catch! We have changed the wording appropriately, throughout the manuscript.

It is surprising that BI-2536 is the top positively correlated gene-compound match as the inhibitor at 5 μ M also inhibits BRD4, inhibition of which produces a very strong phenotype, a point worth discussing.

Based on the similarity of cell images and image-based profiles, we find that:

- BI-2536 correlates much more strongly with PLK1 than to BRD4 (CRISPR data).
- Given the role of PLK1 in mitosis, inhibition is likely to lead to mitotic arrest and apoptosis earlier, and/or BI-2536's PLK1 inhibition effect is likely stronger than the BRD4 effect.
- PFI-1, which is annotated as a BRD4-only inhibitor, does not produce a phenotype that is similar to BI-2536. Coupled with the fact that the PFI-1 phenotype and BRD4 knockout phenotype do not look dissimilar from negative control, we believe that the BRD4 phenotype is not visible with this assay under these experimental conditions.
- There is a strong correlation between BI-2536 treatment and BRD4 overexpression, although it is a relatively non-specific phenotype: both cause cell death. Overexpression of PLK1 does not produce a phenotype distinguishable from the negative control.

We have included the following text in **red** in the manuscript and have also added a new figure, Extended Data Fig. 2 (included below).

For example, the top positively correlated gene-compound match in U2OS cells is the PLK1 inhibitor compound BI-2536 matched with CRISPR against PLK1 (**Extended Data Fig. 2**).

Extended Data Fig. 2: **Similarity of perturbation impact across modalities for genes and compounds related to the compound BI-2536.** Treatment of U2OS cells with BRD4 inhibitors BI-2536 (multi-target, including PLK1 and BRD4) and PFI-1 (BRD4-specific) is shown (top row; all images are composites of max intensity across all five imaging channels). BI-2536 causes cell death, and this phenotype is mimicked by PLK1 knockout (middle row). In contrast, BRD4 knockout fails to produce a distinct phenotype, death-related or otherwise, as is the case for the BRD4-specific inhibitor PFI-1 (middle column, top and middle row); both profiles are quantitatively similar to negative controls. This implies that BRD4 inhibition has a limited phenotypic impact in this assay under these experimental conditions, allowing the PLK1-inhibiting phenotype of BI-2536 to dominate the profile. BRD4 overexpression, on the other hand, also induces cell death (bottom row) and a profile strongly similar to BI-2536, which could indicate that BRD4 overexpression yields a dominant negative phenotype. Overexpression of PLK1 produces a phenotype that is not distinguishable from negative control. Negative controls for compounds, CRISPR, and ORF reagents are included. These are representative images from one of the four replicate wells of each treatment in the dataset. Wells were sampled from the longer time point for each perturbation modality (Supplementary Table 1).

Final Decision Letter:

Dear Anne,

I am pleased to inform you that your Resource, "Three million images and morphological profiles of cells treated with matched chemical and genetic perturbations", has now been accepted for publication in Nature Methods. The received and accepted dates will be May 23, 2023 and March 11, 2024. This note is intended to let you know what to expect from us over the next month or so, and to let you know where to address any further questions.

Over the next few weeks, your paper will be copyedited to ensure that it conforms to Nature Methods style. Once your paper is typeset, you will receive an email with a link to choose the appropriate publishing options for your paper and our Author Services team will be in touch regarding any additional information that may be required. It is extremely important that you let us know now whether you will be difficult to contact over the next month. If this is the case, we ask that you send us the contact information (email, phone and fax) of someone who will be able to check the proofs and deal with any last-minute problems.

Please note that *Nature Methods* is a Transformative Journal (TJ). Authors may publish their research with us through the traditional subscription access route or make their paper immediately open access through payment of an article-processing charge (APC). Authors will not be required to make a final decision about access to their article until it has been accepted. Find out more about Transformative Journals

You may wish to make your media relations office aware of your accepted publication, in case they consider it appropriate to organize some internal or external publicity. Once your paper has been scheduled you will receive an email confirming the publication details. This is normally 3-4 working days in advance of publication. If you need additional notice of the date and time of publication,

please let the production team know when you receive the proof of your article to ensure there is sufficient time to coordinate. Further information on our embargo policies can be found here: <https://www.nature.com/authors/policies/embargo.html>

If you are active on Twitter/X, please e-mail me your and your coauthors' handles so that we may tag you when the paper is published.

Best regards,
Rita

Rita Strack, Ph.D.
Senior Editor
Nature Methods